# PHI-S: Distribution Balancing for Agglomerative Models

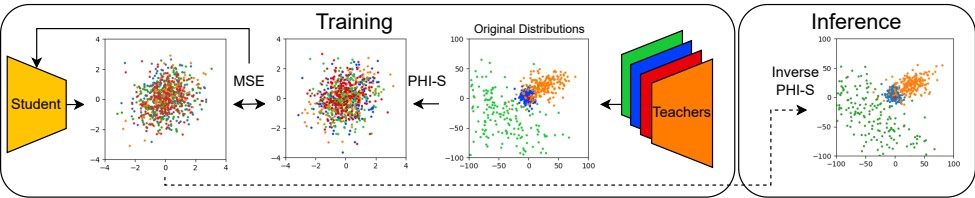

Figure 1: Illustration of the modified agglomerative model training procedure. Instead of the student model learning to match the original teacher distributions, it learns to match the normalized distributions (our proposed PHI-S is shown). We show the real distributions for DFN CLIP, DINOv2, SigLIP, and SAM by projecting them down to 2D using PCA. In the original space, the variance of DFN CLIP is so small that it appears as a single point. During inference, we can estimate the original teacher distributions using the inverse normalization process on the student predictions.

## Abstract

Various visual foundation models have distinct strengths and weaknesses, both of which can be improved through heterogeneous multi-teacher knowledge distillation without labels, termed "agglomerative models." We build upon this body of work by studying the effect of the teachers' activation statistics, particularly the impact of the loss function on the resulting student model quality. We explore a standard toolkit of statistical normalization techniques to better align the different distributions and assess their effects. Further, we examine the impact on downstream teacher-matching metrics, which motivates the use of Hadamard matrices. With these matrices, we demonstrate useful properties, showing how they can be used for isotropic standardization, where each dimension of a multivariate distribution is standardized using the same scale. We call this technique "PHI Standardization" (PHI-S) and empirically demonstrate that it produces the best student model across the suite of methods studied.

## 1 Introduction

A body of work recently emerged on the topic of agglomerative models (Ranzinger et al. (2024)), which is fusing multiple heterogeneous visual foundation models (Awais et al. (2023)) into a single model via multi-teacher knowledge distillation (Hinton et al. (2015); Zuchniak (2023)) without labels. Starting with AM-RADIO (Ranzinger et al. (2024)), and followed by Theia (Shang et al. (2024)), and UNIC (Sariyildiz et al. (2024)). Theia and UNIC apply feature standardization to the teacher output, and demonstrate the importance of the approach.

While knowledge distillation has a large body of literature ( Buciluǎ et al. (2006); Ahn et al. (2019); Heo et al. (2019); Huang & Wang (2017); Romero et al. (2014); Sun et al. (2021); Wei et al. (2022a); Zagoruyko & Komodakis (2017)), agglomerative models - dealing with multiple teachers coming from different modeling domains (e.g. vision-language contrastive Radford et al. (2021), self-supervised learning Oquab et al. (2023); Zhou et al. (2022); Assran et al. (2023), and segmentation Kirillov et al. (2023)) without ground truth labels - was new territory. In AM-RADIO, the authors chose DFN CLIP (Fang et al. (2023)), DINOv2-g-reg (Darcet et al. (2023)), and SAM (Kirillov et al. (2023)) as their teacher models. While the authors studied loss balancing between the

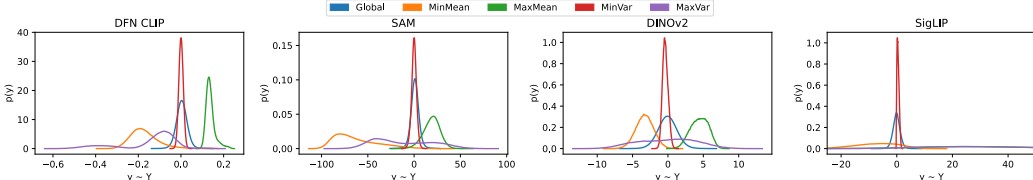

Figure 2: Teacher activation histograms. We show the global histogram, as well as the histograms for the channels associated with the minimum mean, maximum mean, minimum variance, and maximum variance. While all being roughly normal, they have very different centers and scales. We provide specific values in table 7 in the appendix.

different teachers to some degree, they landed on a simple balancing strategy which was to apply the same weight to each teacher, both for summary and feature losses, and to use a linear combination of Cosine and Smooth-L1 (Girshick (2015)) objectives for feature matching.

In this work we study whether the choice of feature distillation loss function in AM-RADIO (equation 3) was an optimal choice. To motivate this, we start by analyzing the feature activation distributions for various teachers in figure 2, and confirm that the distributions have very different variances. Notably, both Mean Squared Error (MSE) and Smooth-L1 are sensitive to variance scale, and thus, left uncontrolled for, each teacher will be implicitly weighted. For example, SAM's distribution has a standard deviation that is $191\times$ larger than that of DFN CLIP. We also note that these distributions aren't a particularity of the training procedure by introducing SigLIP (Zhai et al. (2023b)) which has gained recent popularity due to its high scores on the OpenCLIP (Ilharco et al. (2021)) leaderboard, as well as strong results within VLLMs (Fang et al. (2024); Li et al. (2024)).

**Main Contributions:**

- We study the distributions of the teachers studied in Ranzinger et al. (2024) (plus SigLIP).

- We employ a statistical toolkit of normalizers, and study the effects on downstream metrics.

- We study the effects of rotation matrices when applying whitening after identifying that the orientation of the normalized teacher distribution may affect the student model's errors.

- We study an application of Hadamard matrices on both whitening and standardization.

- In the case of standardization, we demonstrate that the Hadamard matrix may be used to produce a distribution that is standardized using a uniform scale across dimensions. We call this normalization method "PHI Standardization" (PHI-S) and demonstrate that it produces the best student models across our evaluation suite.

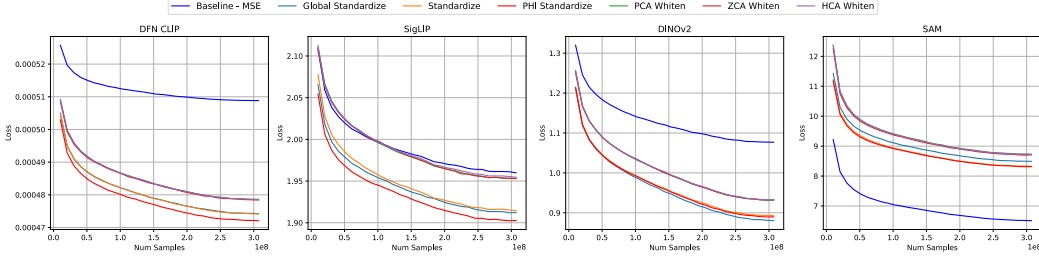

Figure 3: The loss curves for each of the four teachers that the ViT-B/16 student is learning to match simultaneously in original teacher space (e.g. denormalized). We emphasize "Baseline - MSE" (Blue) and "PHI Standardize" (PHI-S, Red) as they generally set the upper and lower bounds.

| Method | Teacher MSE | Classif-ication | Segment-ation | SAM COCO | LLaVA 1.5 | Probe 3D | Avg | Avg No MSE | Avg No COCO | Avg No MSE/COCO |
|---|---|---|---|---|---|---|---|---|---|---|
| | | | | **Baselines** | | | | | | |
| MSE | 6.25 | 10.00 | 10.00 | **1.00** | 9.75 | 10.00 | 7.83 | 8.15 | 9.20 | 9.94 |
| Cosine | 10.00 | **1.00** | **2.00** | 8.00 | 6.63 | 7.25 | 5.81 | 4.98 | 5.38 | 4.22 |
| Hyb MSE | 5.50 | 9.00 | 3.00 | 2.00 | 7.63 | 7.50 | 5.77 | 5.83 | 6.53 | 6.78 |
| Hyb SmL1 | 7.00 | 3.00 | 4.50 | 7.00 | 4.38 | 6.00 | 5.31 | 4.98 | 4.98 | 4.47 |
| | | | | **Standardization** | | | | | | |
| Global Stdze | 2.75 | 3.00 | 4.50 | 5.00 | 3.13 | 4.50 | 3.81 | 4.03 | 3.58 | 3.78 |
| Standardize | 2.75 | 5.00 | 7.50 | 3.00 | **2.75** | 4.25 | 4.21 | 4.50 | 4.45 | 4.88 |
| **PHI-S** | **2.00** | 3.00 | **2.00** | 4.00 | 3.50 | **3.00** | **2.92** | **3.10** | **2.70** | **2.88** |
| | | | | **Whitening** | | | | | | |
| PCA-W | 6.50 | 7.50 | 7.50 | 6.00 | 6.50 | 3.75 | 6.29 | 6.25 | 6.35 | 6.31 |
| ZCA | 5.50 | 7.50 | 8.00 | 9.00 | 4.75 | 4.00 | 6.46 | 6.65 | 5.95 | 6.06 |
| HCA | 6.75 | 6.00 | 6.00 | 10.00 | 6.00 | 4.75 | 6.58 | 6.55 | 5.90 | 5.69 |

Table 1: Average ordinal rank between methods (1 is best, 10 is worst) across the benchmark suite for ViT-B/16. We observe that the standardization techniques work the best, with PHI-S being the strongest normalization method studied. The raw benchmark scores are provided in appendix A.9.2.

## 2 METHOD

The goal of the agglomerative student model is to produce activations $\mathbf{x}^{(k)}$ that match the teacher activations $\mathbf{y}^{(k)} \in \mathbf{Y}^{(k)}$ as closely as possible for each teacher $k \in T$, and the loss is (usually) linearly aggregated using weights $\alpha^{(k)}$. Finding these $\alpha^{(k)}$ is difficult due to the size of the design space, so current methods typically default to $\alpha^{(k)} = 1$ and focus on conditioning $\mathbf{Y}^{(k)}$ to handle distributional differences. For simplicity, we drop the $\cdot^{(k)}$ superscript as the same type of normalization is applied for every teacher, and each teacher has independent normalization parameters. Throughout this paper, we refer to Var $[\mathbf{Z}]$ as the diagonal of the covariance matrix $\mathbf{\Sigma}[\mathbf{Z}]$ for distribution $\mathbf{Z}$.

### 2.1 BASELINE

We start with the MSE (mean squared error) loss serving as the baseline for feature matching:

$$L_{\mathrm{mse}}(\mathbf{x}, \mathbf{y}) = \frac{1}{N} \sum_{n=1}^{N} (\mathbf{x}_n - \mathbf{y}_n)^2 \tag{1}$$

Because AM-RADIO (Ranzinger et al. (2024)) doesn't use MSE as their loss function, but rather a hybrid cosine + Smooth-L1 loss, we also consider a few of these variants. For example, the vanilla cosine distance loss, which is identical to what AM-RADIO uses for the summary loss. While we expect this to do poorly on the task of exactly matching the teacher distribution (due to magnitude invariance), it's not clear how this will affect the downstream tasks, so we include it.

$$L_{\mathrm{cos}}(\mathbf{x}, \mathbf{y}) = \frac{1}{N} \sum_{n=1}^{N} \left(1 - \frac{\mathbf{x}^{\mathsf{T}} \mathbf{y}}{\|\mathbf{x}\| \|\mathbf{y}\|}\right) \tag{2}$$

We also consider the exact loss function proposed in AM-RADIO which is a hybrid of cosine distance and smooth-L1:

$$L_{\mathrm{hyb\text{-}sml1}}(\mathbf{x}, \mathbf{y}) = \beta \cdot L_{\mathrm{cos}}(\mathbf{x}, \mathbf{y}) + (1 - \beta) \cdot \mathrm{SmoothL1}(\mathbf{x}, \mathbf{y}) \tag{3}$$

For completeness, we ablate whether MSE vs Smooth-L1 has an effect on the evaluation criteria:

$$L_{\mathrm{hyb\text{-}mse}}(\mathbf{x}, \mathbf{y}) = \beta \cdot L_{\mathrm{cos}}(\mathbf{x}, \mathbf{y}) + (1 - \beta) \cdot L_{\mathrm{mse}}(\mathbf{x}, \mathbf{y}) \tag{4}$$

In AM-RADIO, the authors used $\beta$ to interpolate between cosine and smooth-L1 loss. Instead of searching the space for the optimal $\beta$, we analyzed the setting they chose ($\beta = 0.9$), and also note that cosine loss corresponds to $\beta = 1.0$ and MSE loss corresponds to $\beta = 0.0$, thus we implicitly study the extremal points of this function interpolation.

| Model | Params (M) | ImageNet1K | | Segmentation (linear) | | Vision-Language (LLaVa-1.5) | | | | SAM |
|---|---|---|---|---|---|---|---|---|---|---|
| | | Zero-shot | k-NN | ADE20k | VOC | GQA | POPE | TextVQA | VQAv2 | COCO |
| AM-RADIO (-H) | 653 | **82.93** | **86.06** | 51.34 | 84.71 | 63.01 | 86.20 | 56.32 | 79.28 | **76.23** |
| PHI-S-RADIO-B | 98 | 74.57 | 82.29 | 48.94 | 84.35 | 63.49 | 86.82 | 57.64 | 79.33 | 73.87 |
| PHI-S-RADIO-L | 320 | 81.01 | 84.93 | **51.47** | **85.49** | 64.29 | 86.86 | 62.48 | 81.10 | 75.06 |

Table 2: Using the PHI Standardization (PHI-S) technique to balance the losses for all of the teachers, we are able to produce ViT-B/16 and ViT-L/16 models using the 3-stage training protocol in appendix A.7 that are competitive with AM-RADIO (ViT-H/16). Notably, our PHI-S-RADIO-L model achieves higher semantic segmentation results, and significantly higher LLaVA-1.5 (Liu et al. (2023)) results. SAM COCO measures the instance mIoU as introduced in Cai et al. (2023).

## 2.2 NORMALIZATION

Instead of balancing the different heads through loss weighting, we can alter the targets themselves. In Wei et al. (2022a), the authors explore this to condition their single teacher's distribution, however, they use the non-invertible LayerNorm operator to rescale the teacher features. Because we want to maintain compatibility for the student to replace the teacher in downstream tasks (by replacing only the vision encoder part of the model), we require the student to still estimate the true teacher distribution. To achieve this, during training, we use an invertible linear mapping $f_k(\cdot)$ such that $T'_k(x) = f_k(T_k(x))$ and $T_k(x) = f_k^{-1}(T'_k(x))$, where the student model learns to match teacher $(T'_k(x))$ for each of the $k$ teachers.

### 2.2.1 STANDARDIZATION

We first consider the simplest case of standardization, which is to use a single scalar $\mu_g$ and std. dev. $\sigma_g$ across the entire feature map. These represent the global statistics of the teacher distribution. In contrast to Wei et al. (2022a), we seek an invertible linear mapping, which excludes LayerNorm. We can, however, estimate the $\mu_{xy}$ and $\sigma_{xy}$ of each position, or, because we want to preserve resolution flexibility, estimate them across all positions and channels, yielding global $\mu_g$ and $\sigma_g$.

Let $\mu_g$ and $\sigma_g$ be the global mean and standard deviation estimate of the teacher distribution. Then

$$L_{\text{gs}}(\mathbf{x}, \mathbf{y}) = L_{\text{mse}}\left(\mathbf{x}, \frac{\mathbf{y} - \mu_g}{\sigma_g}\right) \tag{5}$$

which we call Global Standardization. We also explore regular multivariate standardization where we normalize each channel of the teacher distribution independently. Let $\mu_c = \mathbb{E}[\mathbf{Y}_c]$ and $\sigma_c = \sqrt{\text{Var}[\mathbf{Y}_c]}$, then standardization is defined as

$$L_s(\mathbf{x}, \mathbf{y}) = L_{\text{mse}}(\mathbf{x}, \mathbf{y}'), \quad y'_c = \frac{y_c - \mu_c}{\sigma_c} \tag{6}$$

### 2.2.2 WHITENING

While standardization normalizes the individual feature variances, it doesn't correct for any covariance between dimensions. We can expand on standardization by also eliminating the covariance between features, called whitening. Let $\mathbf{\Sigma}[\mathbf{Y}]$ be the covariance matrix for $\mathbf{Y}$ where $\mathbf{y} \sim \mathbf{Y}$. Following Kessy et al. (2018), we want to find the $\mathbf{W}$ in

$$\mathbf{z} = \mathbf{W}\mathbf{y} \tag{7}$$

with $\mathbf{z} \sim \mathbf{Z}$ and $\mathbf{\Sigma}[\mathbf{Z}] = \mathbf{I}$. $\mathbf{W} = \mathbf{\Sigma}[\mathbf{Y}]^{-\frac{1}{2}}$ is one such valid matrix, called ZCA Whitening (Bell & Sejnowski (1997)), and takes the form

$$\mathbf{y}' = \mathbf{\Sigma}[\mathbf{Y}]^{-\frac{1}{2}}(\mathbf{y} - \boldsymbol{\mu}) \tag{8}$$

Each feature in $\boldsymbol{\Sigma}\left[\mathbf{Y}'\right]$ is linearly independent and has uniform scale. And so $L_w(\mathbf{x}, \mathbf{y}) = L_{\text{mse}}\left(\mathbf{x}, \mathbf{W}\mathbf{y} - \boldsymbol{\mu}\right)$ for any whitening method $w$. $\mathbf{y}$ and $\mathbf{y}'$ are related to each other as

$$\mathbf{y} = \boldsymbol{\Sigma}\left[\mathbf{Y}\right]^{\frac{1}{2}}\mathbf{y}' + \boldsymbol{\mu} \tag{9}$$

### 2.2.3 ESTIMATION ERRORS

Following the whitening notation of Kessy et al. (2018), given some orthogonal matrix $\mathbf{Q}$, then $\mathbf{Q}\mathbf{W}$ is also a valid whitening matrix, as $\mathbf{Q}^\intercal\mathbf{Q} = \mathbf{I}$, therefore $(\mathbf{Q}\mathbf{W})^\intercal\mathbf{Q}\mathbf{W} = \boldsymbol{\Sigma}\left[\mathbf{Y}\right]^{-1}$. Kessy et al. (2018) then demonstrate the properties of certain choices of $\mathbf{Q}$, and we focus on PCA Whitening (PCA-W) and ZCA in this paper. With

$$\boldsymbol{\Sigma}\left[\mathbf{Y}\right] = \mathbf{U}\boldsymbol{\Lambda}\mathbf{U}^\intercal \tag{10}$$

$$\mathbf{W}_{\text{pca-w}} = \mathbf{Q}_{\text{pca-w}}\boldsymbol{\Lambda}^{-\frac{1}{2}}\mathbf{U}^\intercal, \quad \mathbf{Q}_{\text{pca-w}} = \mathbf{I} \tag{11}$$

$$\mathbf{W}_{\text{zca}} = \mathbf{Q}_{\text{zca}}\boldsymbol{\Lambda}^{-\frac{1}{2}}\mathbf{U}^\intercal, \quad \mathbf{Q}_{\text{zca}} = \mathbf{U} \tag{12}$$

where $\mathbf{U}$ and $\boldsymbol{\Lambda}$ are the eigenvectors and eigenvalues for the covariance matrix of $\mathbf{Y}$ respectively. $\boldsymbol{\Lambda} = \text{diag-embed}\left(\lambda_1, ..., \lambda_C\right)$ where diag-embed$(\cdot)$ forms a diagonal matrix with the vector argument along the diagonal. From equation 9, an issue naturally arises, which is the estimation error of our student network. Let $\boldsymbol{\epsilon} \in \mathbb{R}^C$ be the estimation error of the student s.t. $\mathbf{y}' = \mathbf{x} + \boldsymbol{\epsilon}$ where $\mathbf{x}$ is the student prediction for a given normalized teacher, forming the exact equality

$$\mathbf{y} = \mathbf{W}^{-1}\left(\mathbf{x} + \boldsymbol{\epsilon}\right) + \boldsymbol{\mu} \tag{13}$$

$$= \mathbf{W}^{-1}\mathbf{x} + \mathbf{W}^{-1}\boldsymbol{\epsilon} + \boldsymbol{\mu} \tag{14}$$

$$\boldsymbol{\epsilon}_{\text{pca-w}} = \mathbf{U}\boldsymbol{\Lambda}^{\frac{1}{2}}\boldsymbol{\epsilon} \tag{15}$$

$$\boldsymbol{\epsilon}_{\text{zca}} = \mathbf{U}\boldsymbol{\Lambda}^{\frac{1}{2}}\mathbf{U}^\intercal\boldsymbol{\epsilon} \tag{16}$$

We can also use the same $\boldsymbol{\epsilon}$ to study standardization (equation 6), taking the form

$$\boldsymbol{\epsilon}_{\text{std}} = \text{diag-embed}\left(\sigma_1, ..., \sigma_C\right)\boldsymbol{\epsilon} \tag{17}$$

As is clear from equations 15, 16 and 17, the choice of normalization will have an impact on the error profile of the model, unless $\boldsymbol{\epsilon}$ counteracts the distortion. We next introduce another $\mathbf{Q}$ not studied in Kessy et al. (2018), which is to use a scaled Hadamard matrix, based on this idea.

### 2.2.4 HADAMARD WHITENING (HCA)

In PCA Whitening, each successive dimension explains the next-largest variance in the data. While this can be a very useful form, we hypothesize that this sort of dimensional loading might not be healthy for a model to learn to match, as effects such as regularization, step size, gradient clipping, etc. may impact the ability of the model to learn each dimension. Instead of ranking the dimensions, we'd like to do the opposite, and find a $\mathbf{Q}$ that explains exactly the same amount of variance irrespective of channel index. It follows that if we could construct an orthogonal basis where each axis captures an identical amount of energy from the diagonal $\boldsymbol{\Lambda}^{-\frac{1}{2}}$ matrix, then we are able to achieve this balance. First, this matrix $\mathbf{R}$ must be orthogonal for it to be a valid $\mathbf{Q}$. Second, in order for the same proportion of the diagonal $\boldsymbol{\Lambda}$ to be captured by each row, then each cell must have the same magnitude. Specifically, $\mathbf{R}_{ij} = \pm\frac{1}{\sqrt{C}}$. These matrices are called Hadamard matrices, and the following is called Sylvester's construction (Sylvester (1867)), valid when $C$ is a power of 2:

$$\mathbf{H}_1 = [1], \quad \mathbf{H}_n = \frac{1}{\sqrt{2}}\begin{bmatrix} \mathbf{H}_{n-1} & \mathbf{H}_{n-1} \\ \mathbf{H}_{n-1} & -\mathbf{H}_{n-1} \end{bmatrix} \tag{18}$$

where $n = \log_2 C + 1$. The only difference from standard Sylvester's construction is the $\frac{1}{\sqrt{2}}$ scaling at each recursive level, which is necessary for all of the vectors to be unit length. Relating back to whitening, we use $\mathbf{H}$ as the rotation matrix $\mathbf{Q}$:

$$\mathbf{W}_{\text{hca}} = \mathbf{Q}_{\text{hca}} \mathbf{\Lambda}^{-\frac{1}{2}} \mathbf{U}^{\mathsf{T}}, \quad \mathbf{Q}_{\text{hca}} = \mathbf{H} \tag{19}$$

and we end up with "Hadamard Whitening" with corresponding error profile:

$$\epsilon_{\text{hada}} = \mathbf{U} \mathbf{\Lambda}^{\frac{1}{2}} \mathbf{H}^{\mathsf{T}} \epsilon \tag{20}$$

This error profile is interesting due to the fact that an error of size $\delta$ along any single dimension $d_1$ will have identical magnitude in the original space as any other dimension $d_2$. We prove this in appendix A.2.1. Further, in appendix A.1.1 we show how some Hadamard matrices whose size is not a power of 2 can be constructed, and how we found an $\mathbf{H}$ for important model sizes such as 768, 1024, 1152, 1280, and 1408.

### 2.2.5 PCA-HADAMARD ISOTROPIC STANDARDIZATION (PHI-S)

A key issue with the previous normalization procedures (aside from global standardization) is that they place disproportionate weight on lower-variance axes. To avoid this distortion, we present the following theorem, and then describe how we apply it as a novel form of standardization:

**Theorem 2.1.** *For any mean-centered normal data distribution* $\mathbf{X} \in \mathbb{R}^{C \times N}$ *with satisfiable Hadamard-matrix dimension* $C$, *there exists an orthogonal transform* $\mathbf{R} \in \mathbb{R}^{C \times C}$ *and scalar* $\alpha \in \mathbb{R}$ *such that* $\text{diag}\left(\mathbf{\Sigma}\left[\alpha \mathbf{R} \mathbf{X}\right]\right) = \mathbf{1}_C$.

*Proof.* Let $\mathbf{\Sigma}\left[\mathbf{X}\right]$ be the covariance matrix of $\mathbf{X}$, and let $\mathbf{\Sigma}\left[\mathbf{X}\right] = \mathbf{U}\mathbf{\Lambda}\mathbf{U}^{\mathsf{T}}$ where $\mathbf{U}$ is an orthogonal matrix, and $\mathbf{\Lambda} = \text{diag-embed}\left(\lambda_1, ..., \lambda_C\right)$, with $\lambda_i$ being the eigenvalues of $\mathbf{\Sigma}\left[\mathbf{X}\right]$. (called PCA).

First, note that $\mathbf{\Sigma}\left[\mathbf{U}^{\mathsf{T}}\mathbf{X}\right] = \mathbf{U}^{\mathsf{T}}\left(\mathbf{U}\mathbf{\Lambda}\mathbf{U}^{\mathsf{T}}\right)\mathbf{U} = \mathbf{\Lambda}$.

Next, let $\mathbf{H} \in \mathbb{R}^{C \times C}$ be a normalized Hadamard matrix, and recall each cell in $\mathbf{H}$ has value $\pm\frac{1}{\sqrt{C}}$. Using the orthogonal transform $\mathbf{H}\mathbf{U}^{\mathsf{T}}$, we get $\mathbf{\Sigma}\left[\mathbf{H}\mathbf{U}^{\mathsf{T}}\mathbf{X}\right] = \mathbf{H}\mathbf{\Lambda}\mathbf{H}^{\mathsf{T}}$.

$$\text{diag}\left(\mathbf{H}\mathbf{\Lambda}\mathbf{H}^{\mathsf{T}}\right)_r = \sum_{i=1}^{C} \lambda_i \left(\pm\frac{1}{\sqrt{C}}\right)^2 = \frac{1}{C}\sum_{i}^{C} \lambda_i \quad \forall r \in C \tag{21}$$

Let

$$\phi = \sqrt{\frac{1}{C}\sum_{i}^{C} \lambda_i} \tag{22}$$

$$\mathbf{\Sigma}\left[\phi^{-1}\mathbf{M}\right] = \phi^{-2}\mathbf{M} = \frac{C}{\sum_{i}^{C} \lambda_i}\mathbf{M} \tag{23}$$

for some matrix $\mathbf{M}$. For $\mathbf{H}\mathbf{\Lambda}\mathbf{H}^{\mathsf{T}}$, we have

$$\text{diag}\left(\mathbf{\Sigma}\left[\phi^{-1}\mathbf{H}\mathbf{U}^{\mathsf{T}}\mathbf{X}\right]\right)_r = \text{diag}\left(\phi^{-2}\mathbf{H}\mathbf{\Lambda}\mathbf{H}^{\mathsf{T}}\right)_r = \frac{C\sum_{i}^{C} \lambda_i}{C\sum_{i}^{C} \lambda_i} = 1 \quad \forall r \in C \tag{24}$$

Therefore

$$\mathbf{R} = \mathbf{H}\mathbf{U}^{\mathsf{T}} \quad \alpha = \phi^{-1} \tag{25}$$

$\square$

For PHI-S, following equation 25 we use

$$\mathbf{W}_{\text{ship}} = \alpha\mathbf{R} \tag{26}$$

Essentially, we first mean center and then rotate the distribution in such a way ($\mathbf{R}$) that the variance along each resulting dimension is identical, allowing us to uniformly scale by $\alpha$ to achieve a standardized distribution.

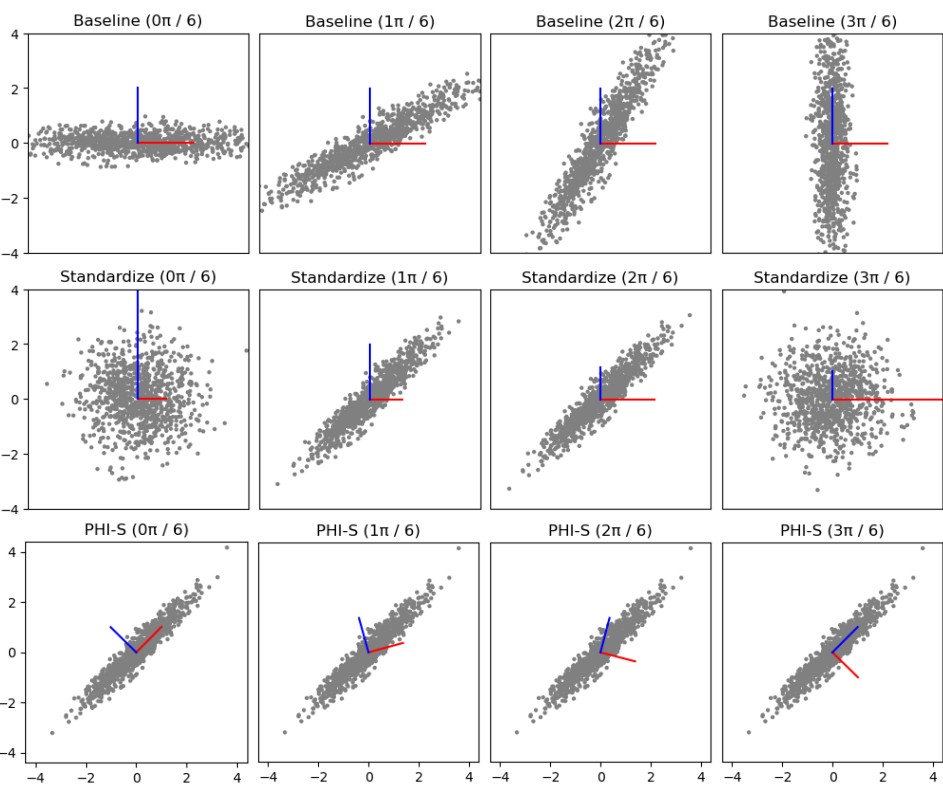

Figure 4: Visualization of how standardization affects the resulting data distribution. We start with the same distribution, and rotate the data by some angle. Regular standardization's effect is directly tied to the distribution orientation. Conversely, PHI-S is invariant to any data rotation, and will produce an identical transform up to sign along each dimension. We can make the sign consistent by negating the rows of $\mathbf{H}$ and $\mathbf{U}$ which have a negative value in the diagonal position. Similarly, regular standardization will distort each dimension (shown with red/blue lines), which will have the effect of reducing the importance of high variance axis-aligned dimensions, and increasing the importance of low-variance dimensions. PHI-S is isotropic, so the change in scale is uniform.

## 3 IMPLEMENTATION DETAILS

We generally follow the procedure outlined in AM-RADIO, however we make some changes that reduce the computational cost of training, which was necessary to cover all of the ablations we studied. Namely, we:

- Add SigLIP as a teacher.
- Train the student model at 256px resolution, and downsample the teacher features to match.
- Train for 300k steps instead of the 600k steps originally proposed.

| Method ↓ | DFN CLIP ($\cdot 1^{-4}$) | SigLIP | DINOv2 | SAM |
|---|---|---|---|---|
| MSE | 5.0883 | 1.9598 | 1.0767 | **6.5082** |
| Cosine | 105.90 | 3.3060 | 1.7980 | 27.9310 |
| Hyb MSE | 7.4930 | 1.9250 | 0.9422 | 7.4580 |
| Hyb SmL1 | 9.8540 | 1.9750 | 0.9112 | 8.6600 |
| Global Stdze | 4.7420 | 1.9120 | **0.8801** | 8.4910 |
| Standardize | 4.7417 | 1.9146 | 0.8928 | 8.3272 |
| **PHI-S (Ours)** | **4.7200** | **1.9010** | 0.8865 | 8.3330 |
| PCA-W | 4.7861 | 1.9534 | 0.9316 | 8.7309 |
| ZCA | 4.7841 | 1.9529 | 0.9321 | 8.7061 |
| HCA (Ours) | 4.7855 | 1.9545 | 0.9326 | 8.7226 |

Table 3: Mean Squared Error for matching the teachers with a ViT-B/16 student using different algorithms. PHI-S does the best job at simultaneously minimizing all teachers.

| | Feature MSE | Classification | Segmentation | SAM COCO | LLaVA 1.5 | Probe 3D | Average | Avg No MSE | Avg No COCO |
|---|---|---|---|---|---|---|---|---|---|
| Baseline MSE | 3.25 | 4.00 | 4.00 | **1.00** | 4.00 | 4.00 | 3.38 | 3.40 | 3.85 |
| Global Stdze | 2.75 | 2.50 | 2.50 | 3.00 | **1.875** | 2.25 | 2.48 | 2.43 | 2.38 |
| Standardize | 2.25 | 2.50 | 2.00 | 2.00 | 2.25 | **1.75** | 2.13 | 2.10 | 2.15 |
| **PHI-S** | **1.75** | **1.00** | **1.50** | 4.00 | **1.875** | 1.75 | **1.98** | 2.03 | **1.58** |

Table 4: Average benchmark ranks for the ViT-L/16 models using the best (and baseline) normalization methods from the ViT-B/16 ablations. PHI-S is even more dominant with the larger model. We provide the raw benchmark scores in appendix A.9.3.

- Split each teacher into their own partition, resulting in each teacher receiving a batch of 256 images, with a total of 1024 images per iteration.
- Initialize from TIMM (Wightman (2019)) "vit_[base,large]_patch16_224" models.

We found that downsampling SAM features degrades their quality, so instead we pad the small image and crop out the features. Further details, and specifically for table 2, are presented in appendix A.7.

## 4 RESULTS

In figure 3 we display our model's ability to estimate the teacher distributions during training. For any of the transforms that project the teacher features into a different space, we apply the inverse operation so that all methods are measured in the original space. As can be seen, "Baseline" is much worse than any other method, and it's intuitive because it allows the relative difference in magnitudes between the different teachers to implicitly weight the loss. SAM has much larger activation variance than any other model, which results in the Baseline model spending most of its energy trying to match SAM. Overall, the PHI Standardization method produces the best results, as it's able to simultaneously beat any other method on DFN CLIP, SigLIP, second best on DINOv2, while remaining competitive on SAM. We show the final MSEs in table 3.

In tables 1 and 4, we display the average benchmark ranks across different benchmarks and methods for ViT-B/16 and ViT-L/16 students, respectively. For LLaVA, we first average the two GQA and TextVQA tasks separately, and then combine them with POPE and VQAv2 to compute the average. This is to prevent overly biasing towards the tasks that have multiple measurements. In both architectures PHI-S produces the best results by achieving the lowest average rank across the suite.

### 4.1 EMPIRICAL ERRORS

In section 2.2.3 we demonstrated how the choice of normalization might have an impact on the errors the student makes when matching the teachers. Particularly, equation 16 is the error profile for ZCA, 15 for PCA-W, 20 for HCA, and 17 for regular standardization. We also have

$$\epsilon_{gs} = \alpha_{gs}^{-1}\epsilon \qquad \epsilon_{\text{ship}} = \alpha_{\text{ship}}^{-1}\epsilon \tag{27}$$

| Method | Normalized | | | | Denormalized | | | |
|---|---|---|---|---|---|---|---|---|
| | DFN CLIP | SigLIP | DINOv2 | SAM | DFN CLIP | SigLIP | DINOv2 | SAM |
| Baseline - MSE | 0.015 | 403.995 | 2.705 | 238.929 | 0.015 | 403.995 | 2.705 | **238.929** |
| Global Stdze | 17.759 | 113.783 | 1.287 | 8.594 | **0.014** | 398.198 | **2.405** | 239.744 |
| Standardize | 0.579 | 0.348 | 0.480 | 0.861 | 0.015 | 406.442 | 2.793 | 240.526 |
| PHI-S | 0.086 | 0.088 | 0.052 | **0.219** | **0.014** | **393.216** | 2.447 | 239.489 |
| PCA-W | 0.416 | 0.339 | 0.830 | 1.634 | 0.015 | 421.195 | 3.179 | 243.610 |
| ZCA | 0.558 | 0.368 | 0.626 | 1.226 | 0.015 | 421.192 | 3.098 | 243.774 |
| HCA | **0.028** | **0.030** | **0.035** | 0.232 | 0.015 | 422.810 | 3.137 | 243.596 |

Table 5: The *range* of the per-channel variances of both the normalized student model errors, as well as the denormalized student errors. A smaller range implies that each channel has a more similar error variance, with $0$ implying that each channel has identical error variance. As theorized, Hadamard and PHI-S have the most uniform variances across the channels, however PHI-S also has the most uniform error variance when projected back into the original (denormalized) space.

for global standardization and PHI-S respectively. We used this error profile to motivate the introduction of Hadamard matrices for whitening in section 2.2.4, as it distributes the error variance equally through all channels of the denormalization projection. In table 5 we display the empirical error variance ranges for each studied method and for each teacher. Intriguigingly, both methods that employ the Hadamard matrix (HCA and PHI-S) have very low variance ranges compared to the other methods. This implies that the student model is making errors of roughly uniform magnitude across all channels. Unfortunately, in the case of HCA, this property isn't borne out in a useful way in the benchmarks (table 1). Table 5 shows that the loss landscape and/or the optimizer are adapting to normalization distortions and baking the non-uniform nature of the variances into the student model. For PHI-S, the student model still has nearly uniform error variance in the normalized space, but also has the lowest (or nearly lowest) range in the denormalized (original) space. This isn't surprising given that a unit change in any dimension of the normalized space has an identical effect as any other dimension, thus there's no incentive to prefer one dimension to another.

# 5 RELATED WORK

**Knowledge Distillation**   We base our work on Ranzinger et al. (2024) which considers a multi-teacher distillation problem without ground-truth labels, and where the targets are the teacher features themselves, instead of estimating e.g. a probability distribution for classification. They build upon extensive literature on knowledge distillation, popularized by Hinton et al. (2015), and then expanded with Kim et al. (2018); Ba & Caruana (2014); Mirzadeh et al. (2019); Beyer et al. (2022) for teacher logit estimation problems. For feature matching, Romero et al. (2014); Huang & Wang (2017); Ahn et al. (2019); Heo et al. (2019); Zagoruyko & Komodakis (2017); Sun et al. (2021); Wei et al. (2022b) study this sub-problem. Specifically, Wei et al. (2022a) discuss the importance of normalizing the teacher feature distribution, which is a notable omission in Ranzinger et al. (2024). Further, in the knowledge distillation domain, the idea of distilling from multiple teachers at once is heavily studied Hinton et al. (2015); Liu et al. (2020); Zuchniak (2023); Yuan et al. (2020); Zhao et al. (2022); Yang et al. (2020); Park & Kwak (2020); You et al. (2017); Lan et al. (2018); Asif et al. (2019); Fukuda et al. (2017). AM-RADIO Ranzinger et al. (2024) differentiates itself from those largely through the lack of a unified target label or distribution, as the teachers aren't even from the same problem domain (e.g. CLIP (Radford et al. (2021)) versus SAM (Kirillov et al. (2023))), and thus will produce very different feature distributions for the same image. Similarly, much of the literature that covers balancing the multi-objective loss relies on having access to ground truth labels (Liu et al. (2020)). Generically, AdaLoss (Hu et al. (2019)) is capable of balancing losses without GT labels by setting the loss weight to be inversely proportional to the approximate expected loss for each term, which AM-RADIO studied but found no significant effect. In Ruder et al. (2017), the authors study domain adaptation where they have multiple classifier teachers from their own domain, and they seek to train a student on a new unlabeled domain, however their method relies on the source and target domains being classification. Concurrently to our work, Shang et al. (2024) introduced the "Theia" model which draws heavily from AM-RADIO including the loss formulation. In their work, the authors chose to use the regular standardization method, a choice which this work explores and demonstrates that it was both a great addition over AM-RADIO, but also not the optimal

choice compared against PHI-S which we propose here. We view the works as complementary, as our study entirely revolves around the design choices in their section 3.2 and AM-RADIO's section 3.4. Recently, UNIC (Sariyildiz et al. (2024)) is also based on AM-RADIO, and employs feature standardization, showing strong positive effects, and UNIT (Zhu et al. (2024)) bases on AM-RADIO employing feature standardization in addition to explicit supervised OCR learning.

**Normalization** The importance of normalization in distillation was identified in Heo et al. (2019), which used BatchNorm. More recently, Wei et al. (2022a) also considered normalized feature matching, however their choice of LayerNorm was non-invertible, and also doesn't de-correlate the different feature dimensions. We aim to preserve the ability of the student to estimate the teacher as in AM-RADIO, so we focus on invertible normalization techniques which allow us to estimate the teacher's true distribution. Liu et al. (2022) argue that normalizing the student and teacher features improves distillation for semantic segmentation as the student otherwise spends most of its energy matching the teacher magnitudes. Intuitively, we expand on this by also observing that controlling the relative magnitudes across teachers is critical. Kessy et al. (2018) provides an overview of different whitening procedures, stressing the fact that there are infinitely many whitening matrices for a given distribution, and focus their attention on the $\mathbf{Q}$ rotation matrix that relates them. Their treatment covers many popular $\mathbf{Q}$ matrices, and we use their work as the foundation for our study. There are also multiple works in the SSL vision domain that deal with distribution properties, such as Barlow Twins (Zbontar et al. (2021)) and VICReg (Bardes et al. (2022)). Their algorithms induce the model to produce regular features, where in contrast, we're forced to deal with arbitrary models that didn't undergo such regularization. In digital signal processing, using the Hadamard matrix to spread energy (to mitigate signal loss errors) is a common practice (Pratt et al. (1969); Kanj et al. (2022)). We study the incorporation of this matrix both as a suitable $\mathbf{Q}$ matrix for rotation during the whitening process, and also in a novel way to derive a scalar normalization factor that standardizes any multivariate distribution with a known Hadamard matrix, which we call PHI Standardization.

## 6 CONCLUSION

Through our experiments, we have conclusively demonstrated that using plain MSE without balancing has a large negative effect on the resulting quality of the student model. Among normalization methods, standardization worked better than whitening, which was an initially surprising result. We hypothesize that the major issue with whitening is that the teacher models aren't producing full rank distributions (appendix, table 6), which makes the normalization factors unstable. Regular standardization is resistant to this because the principal components of the distribution are spread out across all of the dimensions, preventing degenerate $\Lambda^{-\frac{1}{2}}$ solutions. We found two novel applications of Hadamard matrices with respect to distribution normalization: HCA and PHI-S. At the ViT-B/16 model scale, we found that isotropic normalization methods (Global Standardize and PHI-S) worked the best, and for ViT-L/16, PHI-S remained the best. On the topic of reconstruction errors, we found no significant result across the whitening methods with respect to downstream metrics, and also found that the per-channel estimation errors were not uniform in general, unless uniform is the optimal choice (HCA and PHI-S), implying that the student model is able to be robust to the potentially high-distortion nature of the different transforms. Overall, PHI-S appears to be the best normalization method studied, and it allowed us to produce ViT-B and ViT-L models that are competitive with the original AM-RADIO (Ranzinger et al. (2024)) ViT-H model.

**Future Work** We've solely explored the use of PHI-S for agglomerative modeling, however it's a general standardization technique when certain assumptions about the data hold such as normality and dimensionality of the distribution. PHI-S could additionally be used to post-hoc standardize the output of existing models. Lastly, an opportunity arises when combining PHI-S with quantization practices (similar to Ashkboos et al. (2024)) in the information retrieval domain as it balances the information across all channels evenly, potentially unlocking higher fidelity quantizers.

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

# A   APPENDIX

## A.1   HADAMARD MATRICES

### A.1.1   CONSTRUCTING HADAMARD MATRICES

Sylvester's construction gives us a convenient way to construct a Hadamard matrix when $C$ is a power of 2. Unfortunately, many of the $C$s we care about aren't such a power. More generally, the Hadamard Conjecture hypothesizes that there exists a valid Hadamard matrix for any $C$ that is divisible by 4. If true, then there are significantly more valid matrices, and in particular, common deep learning choices will be a multiple of 4. While not proven in general, the literature has found a way to construct many non-power-of-2 sized matrices using some of the following rules:

- If $\mathbf{H}_n$ and $\mathbf{H}_m$ are Hadamard matrices, then $\mathbf{H}_n \bigotimes \mathbf{H}_m$ is also a Hadamard matrix.
- If $3 \equiv q^k \mod 4$ for some prime $q$ and integer $k > 0$, then we can use Paley's first construction Paley (1933) to produce a Hadamard matrix of size $q + 1$.
- If $1 \equiv q^k \mod 4$ for some prime $q$ and integer $k > 0$, then we can use Paley's second construction to produce a Hadamard matrix of size $2(q + 1)$.

where $\bigotimes$ is the Kronecker product. For our purposes, there are common feature dimensions that we want to be able to produce:

- ViT-B: 768 $[\mathbf{S}(2) \bigotimes \mathbf{P}_1(384))]$
- ViT-L: 1024 $[\mathbf{S}(1024)]$
- SigLIP-L: 1152 $[\mathbf{S}(32) \bigotimes \mathbf{P}_2(36)]$
- ViT-H: 1280 $[\mathbf{S}(64) \bigotimes \mathbf{P}_1(20)]$
- ViT-g: 1408 $[\mathbf{S}(32) \bigotimes \mathbf{P}_1(44)]$

Where $\mathbf{P}_i(x)$ is a Paley construction $i$ of size $x$, and $\mathbf{S}(x)$ is a Sylvester construction of size $x$. In the case of Sylvester, we're referring to when $2^k = x$ for some $k \in \mathbb{N}_0$. For $\mathbf{P}_1(384)$, we have the prime $q = 383$, which $3 \equiv 383^1 \mod 4$. For 1280, we can use (possibly among other options) $\mathbf{P}_1(1280)$ as we have $q = 1279$, and thus $3 \equiv 1279^1 \mod 4$, or the compound version shown above. Finally, for $P(44)$ we have $q = 43$ and $3 \equiv 43^1 \mod 4$. So, by some stroke of luck, we have known constructions of Hadamard matrices for the major ViT widths. There are even more methods for constructing these matrices, and at the time of this writing, the smallest unknown Hadamard matrix is 668. While not exhaustive, for our purposes, the Sylvester and Paley constructions were sufficient to cover the models we studied.

### A.1.2   USING HADAMARD MATRICES FOR NOISE SUPPRESSION / QUANTIZATION

While unrelated to our work of using Hadamard matrices to perform statistical normalization, the recently proposed QuaRot Ashkboos et al. (2024) finds a different application of this structured matrix to eliminate activation outliers, making low-bit quantization much more effective.

## A.2   HADAMARD WHITENING

### A.2.1   PROOF OF HCA UNIFORM ERROR PROFILE

Referring to equation 20:

$$\epsilon_{\text{hada}} = \mathbf{U}\mathbf{\Lambda}^{\frac{1}{2}}\mathbf{H}^{\mathsf{T}}\epsilon \qquad\qquad \text{(20 revisited)}$$

we demonstrate that each column of $\mathbf{U}\mathbf{\Lambda}^{\frac{1}{2}}\mathbf{H}^{\mathsf{T}}$ has identical magnitude, and further, that an error step of size $\delta$ along any single dimension has identical magnitude in the original space.

$$\mathbf{\Lambda}^{\frac{1}{2}}\mathbf{H}^{\mathsf{T}} = \frac{1}{\sqrt{C}}\begin{bmatrix} \pm\sqrt{\lambda_1} & \dots & \pm\sqrt{\lambda_1} \\ \pm\sqrt{\lambda_2} & \dots & \pm\sqrt{\lambda_2} \\ \vdots & \ddots & \vdots \\ \pm\sqrt{\lambda_C} & \dots & \pm\sqrt{\lambda_C} \end{bmatrix} \tag{28}$$

$$\left\|\mathbf{\Lambda}^{\frac{1}{2}}\mathbf{H}^{\mathsf{T}}\right\|_{[:,j]} = \sqrt{\sum_{c=1}^{C}\frac{\lambda_c}{C}} \quad \forall j \in C \tag{29}$$

where $\|\cdot\|_{[:,j]}$ denotes the norm of column $j$. Equation 29 shows that each column vector has an identical magnitude. Because orthogonal transforms are magnitude preserving, we also get

$$\left\|\mathbf{U}\mathbf{\Lambda}^{\frac{1}{2}}\mathbf{H}^{\mathsf{T}}\right\|_{[:,j]} = \sqrt{\sum_{c=1}^{C}\frac{\lambda_c}{C}} \quad \forall j \in C \tag{30}$$

In particular, this means that for some $\mathbf{\Delta}_r \in \delta\left[\pm\mathbb{1}_{[r=1]}, \dots, \pm\mathbb{1}_{[r=C]}\right]^{\mathsf{T}}$ with $\mathbb{1}_{r=x}$ representing the Kronecker delta for whether $r = x$ and $\delta \in \mathbb{R}_+$ (e.g. $\mathbf{\Delta}_r$ is a one-hot column vector with a $\pm 1$ at position $r$ multiplied by some positive real $\delta$), then

$$\left\|\left(\mathbf{U}\mathbf{\Lambda}^{\frac{1}{2}}\mathbf{H}^{\mathsf{T}}\right)\mathbf{\Delta}_r\right\| = \delta\sqrt{\sum_{c=1}^{C}\frac{\lambda_c}{C}} \quad \forall r \in C \tag{31}$$

In words, an error step of size $\delta$ along any single axis $r$ in whitened space will be scaled by

$$\sqrt{\frac{1}{C}\sum_{c=1}^{C}\lambda_c} \tag{32}$$

when projecting back into the original space. So each dimension being learned by the student has the same magnitude of effect in the original teacher space. Our hypothesis is that this should improve learning dynamics as there is no implicitly more important dimension to match than any other, compared with PCA-W which places the most importance on the first dimension, and so on. Note that an arbitrary error vector of magnitude $\delta$ does not have this property since $\mathbf{U}\mathbf{\Lambda}^{\frac{1}{2}}\mathbf{H}^{\mathsf{T}}$ is not orthogonal in general.

Incidentally, equation 32 is identical to equation 22 which is the radius of the denormalized unit error circle for PHI-S. This means that at any $\mathbf{\Delta}_r$ with $\delta = 1$, the error magnitude is identical between the two normalization methods. We visualize this when $C = 2$ in figure 8 by looking at where the blue and purple curves intersect.

### A.2.2 VISUALIZING DISTRIBUTIONS

In figure 5 we show how the various normalization transforms change the target distribution, and also how the transforms affect the errors coming from the student model. For the whitening transforms, the choice of $\mathbf{Q}$ matrix has an impact on the relationship between errors of the same magnitude (e.g. fixed radius) in the learned distribution versus the denormalized distribution. Using the Hadamard matrix as $\mathbf{Q}$ is the only choice that doesn't place extra error on a particular learned dimension.

In figure 8 we display the radius of the denormalized error circle. An interesting property of standardization becomes apparent, which is that the error magnitude of standardization is bounded between PCA-W and PHI-S, with equality at $\mathbf{\Sigma}\left[\mathbf{Y}\right] = \mathbf{\Lambda}$ for the former and $\mathrm{diag}\left(\mathbf{\Sigma}\left[\mathbf{Y}\right]\right) = \phi_{\mathrm{phi\text{-}s}}\mathbf{I}$ for the latter. One hypothesis for why the standardization transforms (Global Standardization, Standardization, PHI-S) work best in our study is because the error amplitudes are "less extreme" than

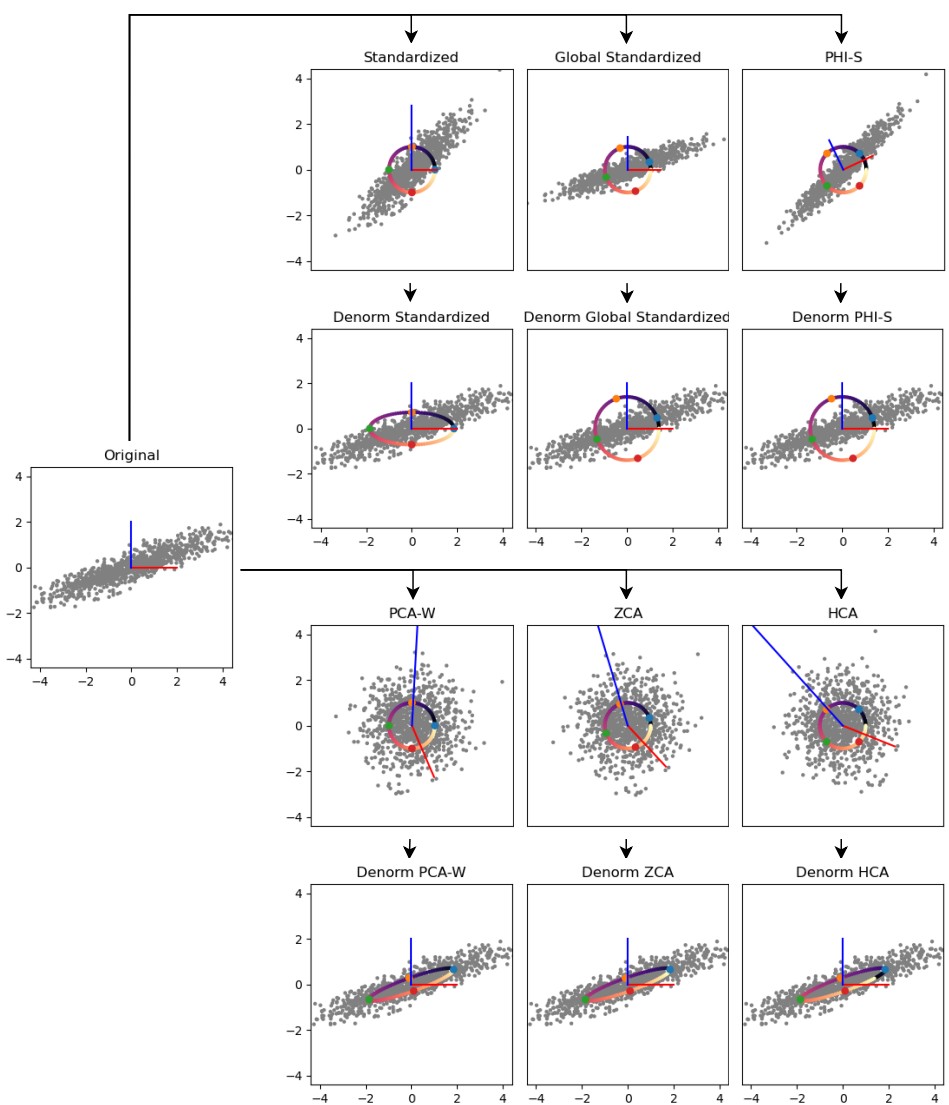

Figure 5: Visualization of normalization procedures. We display two axis lines in red and blue. In the original space, they're both 2 units long, and aligned with the plot coordinate system. We also display an "error circle" which is a unit circle in the normalized coordinate system. For the three whitening transforms you can see how they only differ by rotation. We also specifically draw colored dots on the error circle corresponding to the extremal points of the error circle when denormalized into an ellipse. PCA-W places the largest error magnitude on the x-axis, given that it's the dimension with largest eigenvalue thus estimation errors along the x dimension will have a much larger impact in the denormalized space. As we show in equation 16, the error for ZCA will be proportional to the original distribution's covariance matrix, and thus, the extremal points are along the eigenvectors of the covariance matrix. Hadamard whitening has the extremal points at $|x_1| = |x_2| = ... = |x_C|$. Global Standardization and PHI-S are both isotropic, which means that there's an infinite number of extremal points, so we instead show the points as they relate to the distribution itself. Similar to ZCA, for Global Standardization these points are along the principal axes. And similar to HCA, the aligned points for PHI-S are when $|x_1| = |x_2| = ... = |x_C|$.

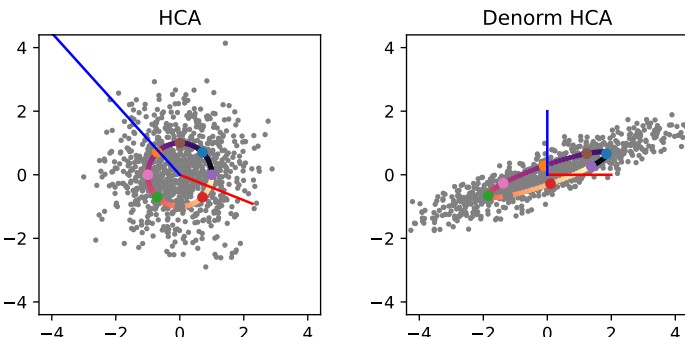

Figure 6: Related to figure 5 and equation 31, we visualize what happens to the one-hot error vectors when projecting back to the original space for HCA. We retain the original $|x| = |y|$ dots, and add the one-hot dots demonstrating how their mapping remains equidistant from the origin relative to each other. In particular, since $\delta = 1$, then $\left\| \left( \mathbf{U}\mathbf{\Lambda}^{\frac{1}{2}}\mathbf{H}^{\mathsf{T}} \right) \mathbf{\Delta}_r \right\| = \sqrt{\frac{1}{C}\sum_c \lambda_c} \approx 1.400892$ for any choice of $r$. For reference, $\mathbf{\Lambda} \approx [3.8356, 0.0894]$.

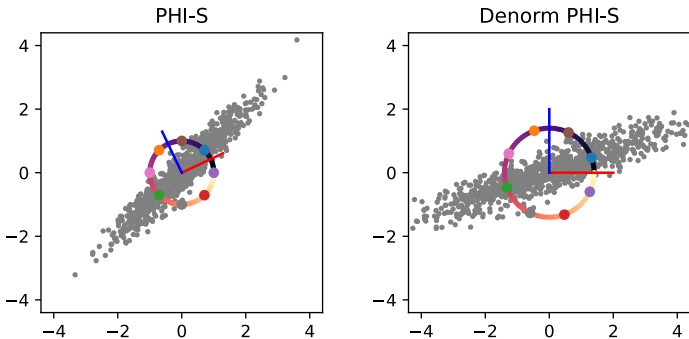

Figure 7: Similar to figure 6 and figure 5, we visualize PHI-S in the normalized and denormalized spaces. This visualizes how equation 27 maintains errors along a circle in both spaces, owing to the isotropic nature of the transform. It also can be seen how the $|x| = |y|$ error dots in normalized space map to the principal directions of the distribution, and also how the one-hot dots capture identical probability density.

whitening in general. With MSE being sensitive to outliers, this property is likely important. Because the whitening methods only differ by an orthogonal transform, their errors are phase shifted relative to each other.

## A.3 TEACHER EFFECTIVE RANKS

We apply the RankMe Garrido et al. (2023) algorithm to a handful of models, including the set of teachers used for training. While it was technically only designed for SSL models (like DINOv2), it may still lend insight into why whitening didn't empirically work well. The results are in table 6, where we show that the effective ranks for all teachers are much smaller than their number of channels. It is also interesting to consider whether agglomerative models work because the teachers aren't effectively full rank, suggesting that we can pack more information from other teachers into a student of equivalent or larger size than the teachers. More investigation is needed to understand why the RADIO models (both AM-RADIO and ours) seem to be lower rank than their counterparts of equivalent size.

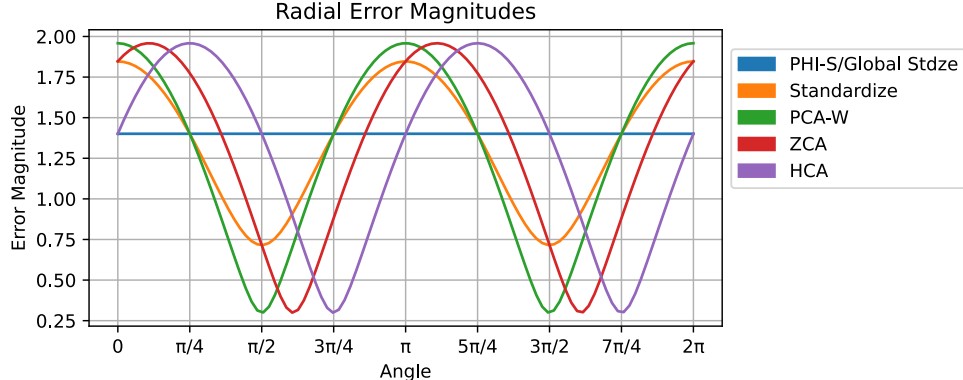

Figure 8: Following from figure 5, we visualize the radius of the denormalized error circle at every angle between 0 and $2\pi$. Because Global Standardization and PHI-S are isotropic, and because the distribution is mean centered (section A.5.2), they scale the error circle uniformly by the same amount. As predicted, for $\theta = z\frac{\pi}{2}$ with $z \in \mathbb{Z}$ (e.g. when $y = 0$ or $x = 0$) we have the same error magnitude for HCA, and also where PHI-S and HCA have identical magnitude. HCA has extremal values at $\theta_{\text{hca}}^{\text{ex}} = z\frac{\pi}{2} + \frac{\pi}{4}$. PCA-W has extremal values at $\theta_{\text{pca-w}}^{\text{ex}} = z\frac{\pi}{2}$. We also have that ZCA will have extremal values $\theta_{\text{pca-w}}^{\text{ex}}(z) \leq \theta_{\text{zca}}^{\text{ex}}(z) \leq \theta_{\text{hca}}^{\text{ex}}(z)$.

| Model | C | RankMe |
|---|---|---|
| DINOv2-b-reg | 768 | 685.52 |
| PHI-S-RADIO-B | 768 | 645.38 |
| DINOv2-l-reg | 1024 | 906.48 |
| PHI-S-RADIO-L | 1024 | 859.23 |
| SigLIP (-L) | 1152 | 910.97 |
| DFN CLIP (-H) | 1280 | 1081.69 |
| SAM (-H) | 1280 | 776.28 |
| AM-RADIO (-H) | 1280 | 1043.84 |
| DINOv2-g-reg | 1536 | 1342.55 |

Table 6: The effective rank estimates for the spatial features of various models using the RankMe Garrido et al. (2023) algorithm. As can be seen, the effective rank is much smaller than $C$, meaning that whitening methods will have a large number of dimensions with very small variance. This likely helps to explain why the whitening methods produced the students with the highest losses.

A.4 TEACHER DISTRIBUTION STATISTICS

In table 7 we show statistics about the distributions of the teachers. We can see that they are not mean centered, and also that their standard deviations are very different, both globally, and per-channel.

| Model | Per Channel | | | | Global | |
|---|---|---|---|---|---|---|
| | Mean | | Std | | Mean | Std |
| | Min | Max | Min | Max | | |
| DFN CLIP | -0.1689 | 0.1385 | 0.0105 | 0.1334 | 0.0049 | 0.0286 |
| SigLIP | -6.8789 | 31.25 | 0.3813 | 21.6875 | 0.0211 | 1.8389 |
| DINOv2 | -3.3945 | 4.293 | 0.3918 | 4.3008 | 0.0055 | 1.3496 |
| SAM | -62.0312 | 19.1719 | 2.6953 | 31.6094 | 1.1475 | 5.4688 |

Table 7: Activation statistics for various teachers. Here we can see that each of the teachers' distributions have very different standard deviations (Global). We can also see that different channels for a given teacher have very different means and standard deviations (Per Channel). Taking SAM as an example: The smallest mean value channel has value $-62.0312$, and largest channel 19.1719. Similarly, the channel with smallest standard deviation has 2.6953, and channel with largest has 31.6094.

### A.5 Additional PHI-S Insights

#### A.5.1 Role of PCA

Because rotations are magnitude preserving (and thus variance preserving), with $\mathbf{\Sigma}\left[\mathbf{Y}\right] = \mathbf{U}\mathbf{\Lambda}\mathbf{U}^{\mathsf{T}}$, then $\mathrm{Tr}\left(\mathbf{\Sigma}\left[\mathbf{Y}\right]\right) = \mathrm{Tr}\left(\mathbf{\Lambda}\right) = \sum_i^C \lambda_i$. This means that the normalization ($\alpha$) derived in equation 25 is a constant with respect to the distribution, invariant to any orthogonal transform that's applied to it. It will always be $\sqrt{\frac{1}{C}\sum_i^C \lambda_i}$. And so, we have that

$$\mathbf{\Sigma}\left[\mathbf{H}\mathbf{Y}\right] = \mathbf{H}\left(\mathbf{U}\mathbf{\Lambda}\mathbf{U}^{\mathsf{T}}\right)\mathbf{H}^{\mathsf{T}} \tag{33}$$

where the $\mathbf{U}$ is preventing the $\mathbf{H}$ from evenly distributing the variance in $\mathbf{\Lambda}$, unless $\mathbf{U} = \mathbf{I}$, or worst case $\mathbf{U} = \mathbf{H}^{\mathsf{T}}$ in which case applying $\mathbf{H}$ would result in $\mathbf{\Lambda}$ variance, the opposite of what we want. So, we don't need PCA to find the scale $\alpha$ to normalize the distribution, but we do need it to find the orthogonal transform $\mathbf{H}\mathbf{U}^{\mathsf{T}}$ which results in a dimensionally balanced distribution.

$$\mathbf{\Sigma}\left[\mathbf{H}\mathbf{U}^{\mathsf{T}}\mathbf{Y}\right] = \left(\mathbf{H}\mathbf{U}^{\mathsf{T}}\right)\left(\mathbf{U}\mathbf{\Lambda}\mathbf{U}^{\mathsf{T}}\right)\left(\mathbf{U}\mathbf{H}^{\mathsf{T}}\right) \tag{34}$$
$$= \mathbf{H}\mathbf{\Lambda}\mathbf{H}^{\mathsf{T}} \tag{35}$$

#### A.5.2 Comparison between Global Standardization and PHI-S

In table 8 we show what the normalization scalars are for each teacher distribution. Because both methods use a single value to rescale the distribution, it's useful to see how they treat the same distribution. Notably, PHI-S uses a larger scale for all of the teacher distributions. It's also worth noting that the difference in scales is not constant across the distributions. Both methods are invariant to the orientation of the distribution, thus these scalars are unique properties of the distribution.

| Model | $\alpha_{\mathrm{gs}}$ | $\alpha_{\mathrm{phi\text{-}s}}$ |
|---:|:---:|:---:|
| DFN CLIP | 35.02 | 41.41 |
| SigLIP | 0.53 | 0.65 |
| DINOv2 | 0.73 | 0.76 |
| SAM | 0.19 | 0.21 |

Table 8: Comparison of scales between global standardization equation 5 and PHI-S standardization equation 26. We get $\alpha_{\mathrm{gs}} = \frac{1}{\sigma_g}$, which is the scaling factor for global standardization.

A natural question arises: Why is $\alpha_{\mathrm{gs}} \neq \alpha_{\mathrm{phi\text{-}s}}$?

Recall that

$$\alpha_{\mathrm{phi\text{-}s}} = \phi^{-1} = \left(\frac{1}{C}\sum_i^C \lambda_i\right)^{-\frac{1}{2}} \tag{22 \& 25 revisited}$$

$$= \left(\frac{1}{C}\mathrm{Tr}\left(\mathbf{\Sigma}\left[\mathbf{Y}\right]\right)\right)^{-\frac{1}{2}} \tag{36}$$

And also how in section A.5.1 we showed that $\alpha_{\mathrm{phi\text{-}s}}$ is invariant to any orthogonal transform on the distribution. For global standardization, we reinterpret the multivariate distribution as univariate, thus we get scalar $\mu_g$ and $\sigma_g$, global mean and global standard deviation respectively. For the multivariate distribution, we have $\boldsymbol{\mu}$, the vector of means for each dimension. We can equivalently write the computation of $\sigma_g$ as

$$\sigma_g = \sqrt{\frac{1}{NC-1}\sum_i^N\sum_c^C \left(y_{i,c} - \mu_g\right)^2} \tag{37}$$

| Model | Mean | Min | Max | # > 0.75 |
|---:|---|---|---|---|
| DFN CLIP | 0.0229 | 0.0000 | 0.9916 | 1 |
| SigLIP | 0.0246 | 0.0001 | 0.7864 | 1 |
| DINOv2 | 0.0203 | 0.0000 | 0.9807 | 1 |
| SAM | 0.0226 | 0.0000 | 0.1128 | 0 |

Table 9: Measuring how aligned the original teacher distribution is with the PHI-S distribution. Refer to section A.5.3 for how this is calculated.

and then for $\phi_{\text{phi-s}}$ we have

$$\phi_{\text{phi-s}} = \sqrt{\frac{1}{N(C-1)} \sum_{i}^{N} \sum_{c}^{C} (y_{i,c} - \mu_c)^2} \tag{38}$$

therefore, when $\mu_c = \mu_g \quad \forall c \in C$, then $\lim_{N \to \infty} \sigma_g = \lim_{N \to \infty} \phi_{\text{phi-s}}$. Meaning that, as long as the mean for each dimension of the distribution is the same, then Global Standardization and PHI-S will arrive at nearly the same scaling constant when N is large, and thus only differ by rotation. A trivial example is when the distribution is already mean centered on every dimension. We show in table 7 that none of the teachers we studied have uniform mean per channel, which is why the methods end up with different scaling constants.

### A.5.3 How similar are the original teacher distributions to the PHI-S distribution?

The main property of PHI-S is that it rotates the distribution in such a way that the standard deviation for each channel is identical, allowing us to standardize the distribution in this rotated space using a single scalar. From equation 25, if the two distributions are aligned, then $\mathbf{H}\mathbf{U}^\mathsf{T} = \mathbf{I}^*$ with $\mathbf{I}^*$ being some permutation of $\mathbf{I}$. We measure the deviation from this ideal by computing $\text{abs}(\mathbf{H}\mathbf{U}^\mathsf{T})$, and then using the Hungarian algorithm Kuhn (1955) to find the best match of basis vectors $\mathbf{U}_{\text{align}}^\mathsf{T}$, and finally calculating statistics on $\text{diag} \circ \text{abs}\left(\mathbf{H}\mathbf{U}_{\text{align}}^\mathsf{T}\right)$, which we show in table 9. We observe that in general, the original distribution is quite unlike that of PHI-S, where at most one basis vector is mostly aligned, but otherwise $\mathbf{H}$ and $\mathbf{U}$ are highly dissimilar.

### A.5.4 Degenerate Rank Distributions

There are additional useful properties for the PHI-S transform, particularly when the original data distribution is not full rank, which is almost certainly the case with deep learning models (table 6, Garrido et al. (2023)). Namely, with the whitening procedures, they will create extreme scale distortions on the zero or negligible eigenvalue dimensions, which can cause the student model to waste too many resources optimizing negligible dimensions. Vanilla standardization also suffers from the same effect, but it may be less aggressive as it's not using PCA which disentangles the dimensions, rather its sensitivity to this problem relies on the orientation of the original distribution. PHI-S, on the other hand, will be well behaved whenever $\text{Rank}(\mathbf{Y}) \geq 1$ because the rotation will place the same amount of variance on every output dimension. We use the definition in Roy & Vetterli (2007) for Rank, the effective rank.

Every normalization method, except for PHI-S and Global Standardization, is vulnerable to when $\text{Rank}(\mathbf{Y}) \leq C$, which we illustrate in the 2-dimensional case:

Let $\mathbf{X} \in \mathbb{R}^{2 \times N}$ be a data distribution with covariance $\mathbf{\Sigma}[\mathbf{X}] = \begin{bmatrix} 1 & 0 \\ 0 & \epsilon \end{bmatrix}$. Because standardization 2.2.1 requires division by the variance in each dimension, then $\lim_{\epsilon \to 0} \sigma_y^{-1} = \infty$. For the whitening methods 2.2.2, the diagonalization of $\mathbf{\Sigma}[\mathbf{X}]$ produces $\mathbf{\Lambda} = \text{diag-embed}(1, \epsilon)$. The whitening methods then require $\mathbf{\Lambda}^{-\frac{1}{2}}$ which again produces a division by 0 for the y-dimension. Because the PHI-S method operates on the mean eigenvalue, it will have $\lim_{\epsilon \to 0} \alpha = \frac{1}{\sqrt{0.5}} = \sqrt{2}$, which is well

defined. While this is a trivial example, the implications are meaningful on real data too, which we show in table 6.

## A.6  NORMALIZATION WITHOUT RUNTIME PENALTY

All of the normalization methods introduce extra computation in the form of mean subtraction and some scaling method. Because the teacher adaptors for our model all end with a $y = \mathbf{W}'\mathbf{x} + \mathbf{b}'$ linear layer, we can modify this layer after training to produce outputs in the original teacher space. Let $\mathbf{y}'$ be the normalized teacher outputs, $\mathbf{y}$ be the original teacher outputs, $\mathbf{x}'^{(n)}$ be the output of the student matching the normalized teacher (at layer $n$), and we seek to produce a $\mathbf{x}^{(n)}$ that approximates the original teacher distribution. With this, we have:

$$\mathbf{x}'^{(n)} = \mathbf{W}'\mathbf{x}'^{(n-1)} + \mathbf{b}' \tag{39}$$

$$\mathbf{x}^{(n)} = \boldsymbol{\Theta}\left(\mathbf{W}'\mathbf{x}^{(n-1)} + \mathbf{b}'\right) + \boldsymbol{\mu} \tag{40}$$

$$= \boldsymbol{\Theta}\mathbf{W}'\mathbf{x}^{(n-1)} + \boldsymbol{\Theta}\mathbf{b}' + \boldsymbol{\mu} \tag{41}$$

$$\mathbf{W} = \boldsymbol{\Theta}\mathbf{W}' \tag{42}$$

$$\mathbf{b} = \boldsymbol{\Theta}\mathbf{b}' + \boldsymbol{\mu} \tag{43}$$

$$\mathbf{x}^{(n)} = \mathbf{W}\mathbf{x}'^{(n-1)} + \mathbf{b} \tag{44}$$

with $\mathbf{W}'$ and $\mathbf{b}'$ being the weights and bias of the final linear layer of the model respectively. $\boldsymbol{\Theta}$ and $\boldsymbol{\mu}$ are the linear correction parameters for the given normalization method.

- **Global Standardize** (2.2.1):
$$\boldsymbol{\Theta} = \mathbf{I}\sigma_g, \quad \boldsymbol{\mu} = \mathbf{1}\mu_g \tag{45}$$

- **Standardize** (2.2.1):
$$\boldsymbol{\Theta} = \text{diag-embed}\left(\sigma_1, ..., \sigma_C\right) \tag{46}$$

- **PCA Whitening** (2.2.2):
$$\boldsymbol{\Theta} = \mathbf{U}\boldsymbol{\Lambda}^{\frac{1}{2}} \tag{47}$$

- **ZCA Whitening** (2.2.2):
$$\boldsymbol{\Theta} = \boldsymbol{\Sigma}\left[\mathbf{Y}\right]^{\frac{1}{2}} \tag{48}$$

- **Hadamard Whitening** (2.2.4):
$$\boldsymbol{\Theta} = \mathbf{U}\boldsymbol{\Lambda}^{\frac{1}{2}}\mathbf{H}^{\mathsf{T}} \tag{49}$$

- **PHI-S** (2.2.5):
$$\boldsymbol{\Theta} = \phi\mathbf{U}\mathbf{H}^{\mathsf{T}} \tag{50}$$

## A.7  IMPLEMENTATION DETAILS

In addition to all of the ablations, we also reported PHI-S-RADIO-B and PHI-S-RADIO-L models (Table 2). To produce these models, we add 2 more training stages on top of that in section 3 as follows:

- Stage 1 - Outlined in section 3 (32 A100 GPUs for 40 hours)

- Stage 2 - Increase the student resolution to 432 and train for 300k more steps (64 A100 GPUs for 64 hours)

- Stage 3 - Add a "high res" set of partitions. Similar to AM-RADIO, we set the batch size to 128 for hi-res while keeping 1024 for low-res. We again train for another 300k steps. (128 A100 GPUs for 68 hours)

| Hyperparameter | Stage 1 | Stage 2 | Stage 3 |
|---|---|---|---|
| Dataset | DC1B | DC1B | DC1B |
| Batch Size | 1024 | 1024 | 1152 |
| GPUs | 32 | 64 | 128 |
| Steps | 300,000 | 300,000 | 300,000 |
| LR | 1e-3 | 1e-3 | 1e-3 |
| LR Schedule | cosine | cosine | cosine |
| Weight Decay | 0.02 | 0.02 | 0.02 |
| Dist Est. Steps | 3,000 | 3,000 | 3,000 |
| Frozen Body Steps | 5,000 | 5,000 | 5,000 |
| Optimizer | LAMB | LAMB | LAMB |

Table 10: Hyperparameter table for the training stages. For each stage, we "restart" the learning rate schedule at $1e-3$. "Dist Est. Steps" describes the number of steps we use at the beginning of the training stage to estimate the teacher data distributions. We reset these estimates for each stage, as the change in resolution may impact these distributions. We also freeze the trunk of the model for "Frozen Body Steps" at the start of each stage to allow for the heads to adjust to the new distributions, and also because these distributions may drastically change early on as the estimates are refined. Particularly, methods that rely on matrix diagonalization can undergo major shifts as PyTorch's implementation of torch.eigh() is not particularly stable under small changes to the covariance matrix. ZCA whitening *is* stable upon small estimate updates, owing to the fact that the $\mathbf{U}^\intercal$ rotation is inverted after rescaling, so any permutation of eigenvectors is also negated. DC1B stands for "DataComp-1B" Gadre et al. (2023), from which we only use the images.

The multi-stage strategy results in 14,080 total GPU hours for the ViT-B/16 model. If we were to instead train stage 3 for 600k steps (AM-RADIO recipe), it would result in 17,408 total GPU hours. Hyperparameters are shown in table 10.

We employ spectral reparametrization Zhai et al. (2023a) for all stages of training. We've found this to be particularly helpful for stage 3 training when dealing with high resolution. In order to encourage the spectral norm to be small, we ensure that weight decay is applied to the rescaling parameter.

## A.8    Loss Distributions By Normalization Method

In figure 9 we show the loss distributions for the core normalization methods we studied. It helps us understand not only the magnitudes of the errors, but also showcases how different normalization methods affect the behavior of outliers. It's very apparent that "Baseline" has uncontrolled magnitudes, with SAM having quite extreme losses, especially relative to DFN CLIP. This is also where we can really see how "Global Standardize" and "PHI-S" differ in behavior, owing to PHI-S equalizing the variance across channels. The purple curve shows how global standardization is still very susceptible to outlier errors. As predicted in section 2.2.3, the methods that use Hadamard matrices (PHI-S and HCA) have the tightest error bounds between channels. Finally, it's also apparent how well PHI-S works for balancing across teachers, as the losses all have the most similar distributions compared against the other methods.

## A.9    Raw Metrics

### A.9.1    Adaptive Balancing

In AM-RADIO, the authors also explore the use of AdaLoss Hu et al. (2019), which sets each loss term to be approximately 1 by dividing the term by the exponential moving average of itself. We explore using this balancing mechanism, both as a standalone (e.g. Baseline + AdaLoss), as well as in conjunction with PHI-S. Table 11 shows the teacher MSEs, and table 12 shows the benchmark ranks with AdaLoss included. In general, AdaLoss places much more weight on the summary losses, resulting in outsized gains in classification tasks at the expense of dense tasks. We also find that AdaLoss+PHI-S is better than AdaLoss alone.

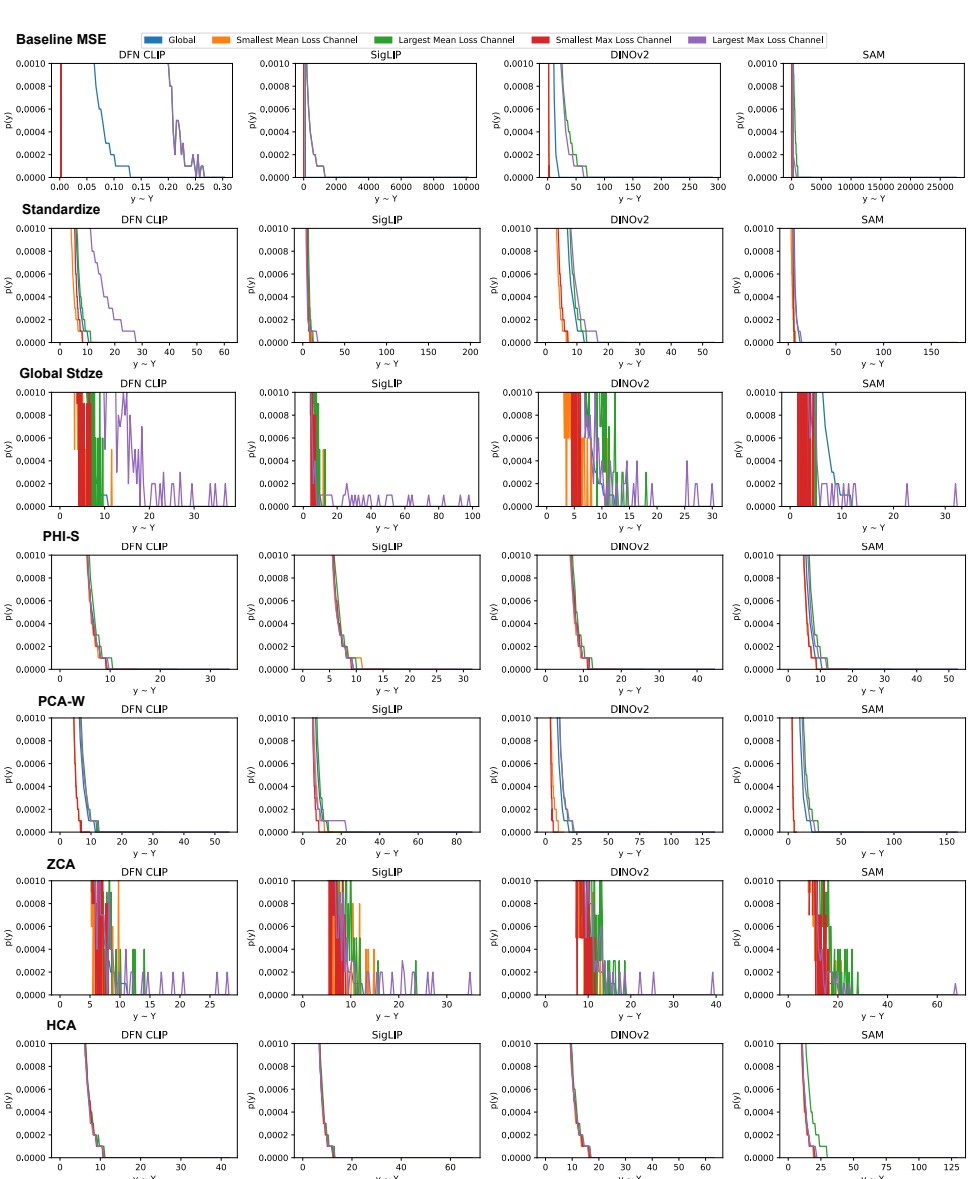

Figure 9: Loss distributions for various normalization methods. The x-axis range is based on the minimum and maximum losses seen for each method over the course of 1,000 samples after training for 100k iterations. The "Largest Max Loss Channel" shows the distribution for the channel that had the highest loss value. It helps us understand how vulnerable our learning process is to outliers. The "Global" curve shows the distribution by combining all of the channels.

| Method ↓ | DFN CLIP ($\cdot 1^{-4}$) | SigLIP | DINOv2 | SAM |
|---|---|---|---|---|
| Ada - MSE | 4.7790 | 1.9260 | 0.9591 | 8.7500 |
| Ada - PHI-S | 4.7750 | 1.9260 | 0.9585 | 8.6960 |

Table 11: Mean Squared Error for matching the teachers with a ViT-B/16 student using AdaLoss, either normally (Ada - MSE), or in conjunction with PHI-S.

| Method | Teacher MSE | Classif-ication | Segment-ation | SAM COCO | LLaVA 1.5 | Probe 3D | Avg | Avg No COCO | Avg No MSE/COCO |
|---|---|---|---|---|---|---|---|---|---|
| | | | | **Baselines** | | | | | |
| MSE | 7.75 | 12.00 | 12.00 | **1.00** | 11.67 | 12.00 | 9.40 | 11.08 | 11.92 |
| Cosine | 12.00 | 3.00 | **2.00** | 10.00 | 7.67 | 7.25 | 6.99 | 6.38 | 4.98 |
| Hyb MSE | 6.00 | 11.00 | 3.00 | 2.00 | 8.83 | 8.25 | 6.51 | 7.42 | 7.77 |
| Hyb SmL1 | 8.00 | 5.00 | 4.50 | 9.00 | 5.17 | 6.00 | 6.28 | 5.73 | 5.17 |
| | | | | **Standardization** | | | | | |
| Global Stdze | 2.75 | 5.00 | 4.50 | 5.00 | 4.33 | 4.50 | 4.35 | 4.22 | 4.58 |
| Standardize | 2.75 | 7.00 | 7.50 | 3.00 | **3.83** | 4.75 | 4.81 | 5.17 | 5.77 |
| **PHI-S** | **2.00** | 5.00 | **2.00** | 4.00 | 4.33 | **3.00** | **3.39** | **3.27** | **3.58** |
| | | | | **Whitening** | | | | | |
| PCA-W | 7.75 | 9.50 | 7.50 | 8.00 | 7.33 | 3.75 | 7.31 | 7.17 | 7.02 |
| ZCA | 6.75 | 9.50 | 8.00 | 11.00 | 5.67 | 4.50 | 7.57 | 6.88 | 6.92 |
| HCA | 8.00 | 8.00 | 6.00 | 12.00 | 7.33 | 5.00 | 7.72 | 6.87 | 6.58 |
| | | | | **AdaLoss** | | | | | |
| MSE | 7.75 | **1.50** | 11.00 | 7.00 | 5.83 | 9.25 | 7.06 | 7.07 | 6.90 |
| PHI-S | 6.25 | **1.50** | 10.00 | 6.00 | 6.00 | 9.75 | 6.58 | 6.70 | 6.81 |

Table 12: Average benchmark ranks across the suite including AdaLoss. For LLaVA, we first average the two GQA and TextVQA tasks separately, and then combine those with POPE and VQAv2 to compute the average. This is to prevent overly biasing towards the tasks that have multiple measurements. We observe that the standardization techniques perform the best, with PHI-S being the strongest normalization method studied. AdaLoss was able to improve over baseline, but is not competitive with the standardization methods. The raw benchmark scores are provided in appendix A.9.2.

## A.9.2 VIT-B/16

In table 13 we show the raw benchmark scores for classification, segmentation, and Probe 3D El Banani et al. (2024). When viewing the raw scores, it's less clear what the ideal method is, if any, aside from it being fairly obvious that the MSE baseline is the worst. We also show the metrics for LLaVA 1.5 integration in 14. It's easiest to see the best performing method by looking at the average ranks across the task suite in table 1, where being consistently strong is more evident. The "Ada -" prefix means that we used AdaLoss.

| | Classification | | Segmentation | | | Probe 3D | | | |
|---|---|---|---|---|---|---|---|---|---|
| Method ↑ | Zero Shot | kNN | ADE20k | VOC | SAM COCO | Depth | Surface Normals | Multi-View | SPair 71k |
| MSE | 56.17 | 71.54 | 42.40 | 78.10 | **71.90** | 77.69 | 55.06 | 47.71 | 33.56 |
| Cosine | 71.44 | 79.74 | 48.01 | **83.39** | 69.42 | 81.77 | 56.46 | 53.53 | 39.59 |
| Hyb MSE | 69.34 | 78.72 | 48.00 | 83.29 | 70.54 | 80.88 | 56.30 | 52.57 | 43.44 |
| Hyb SmL1 | 71.19 | 79.49 | 48.23 | 82.82 | 69.53 | **82.14** | 56.43 | 53.69 | 40.45 |
| Global Stdze | 70.91 | 79.51 | 47.89 | 83.07 | 69.75 | 82.02 | 57.02 | 54.13 | 42.53 |
| Standardize | 70.51 | 79.35 | 47.87 | 82.79 | 70.22 | 80.44 | 56.48 | **54.65** | **45.27** |
| PHI-S | 70.73 | 79.53 | **48.63** | 83.09 | 69.89 | 81.89 | 56.79 | 54.49 | 43.92 |
| PCA-W | 70.23 | 79.30 | 47.58 | 82.96 | 69.55 | 81.88 | 56.71 | 54.42 | 44.24 |
| ZCA | 70.38 | 79.28 | 47.83 | 82.80 | 69.37 | 81.43 | **57.23** | 54.49 | 43.35 |
| HCA | 70.47 | 79.33 | 47.84 | 82.99 | 69.19 | 81.61 | 57.07 | 54.35 | 43.14 |
| Ada - MSE | **72.89** | 79.85 | 47.24 | 82.53 | 69.60 | 81.57 | 56.33 | 51.86 | 36.85 |
| Ada - PHI-S | 72.73 | **80.03** | 47.41 | 82.72 | 69.74 | 81.72 | 55.49 | 51.28 | 36.46 |

Table 13: **ViT-B/16** - Classification accuracy using both Zero Shot (DFN CLIP text encoder) and kNN. ADE20k and VOC are semantic segmentation linear probe results using 512px resolution (see Ranzinger et al. (2024) for details), and SAM COCO instance segmentation, also defined in AM-RADIO. We also show the Probe 3D El Banani et al. (2024) metrics as also reported in AM-RADIO.

## A.9.3 VIT-L/16

In table 15 we show the MSE for our ViT-L/16 trained student model. Similar to the ViT-B/16 metrics, PHI-S does the best job of simultaneously minimizing all of the teacher errors. We also provide the raw benchmark scores in tables 16 and 17.

| Method ↑ | GQA | | TextVQA | | POPE | VQAv2 |
|---|---|---|---|---|---|---|
| | Val | TestDev | Tokens | No Tokens | | |
| MSE | 67.35 | 59.51 | 47.31 | 15.06 | 85.16 | 72.21 |
| Cosine | 70.02 | 61.82 | 50.24 | 24.13 | 84.78 | 76.14 |
| Hyb MSE | 69.86 | 61.96 | 50.15 | 23.53 | 85.19 | 75.94 |
| Hyb SmL1 | 70.03 | 62.35 | 50.19 | 23.90 | 85.74 | 76.17 |
| Global Stdze | 70.10 | 62.28 | 50.31 | 22.55 | 85.88 | 76.21 |
| Standardize | 70.04 | 62.16 | 50.28 | 24.20 | **85.94** | 76.20 |
| PHI-S | **70.20** | **62.55** | 50.25 | 23.28 | 85.52 | **76.30** |
| PCA-W | 69.85 | 62.01 | 50.48 | 24.14 | 85.43 | 75.93 |
| ZCA | 69.98 | 62.37 | 50.11 | 24.63 | 85.80 | 76.02 |
| HCA | 69.95 | 61.79 | 49.92 | 24.79 | 85.61 | 76.07 |
| Ada - MSE | 69.75 | 62.31 | 50.82 | **26.63** | 85.06 | 76.09 |
| Ada - PHI-S | 69.76 | 61.90 | **50.90** | 25.81 | 85.54 | 76.03 |

Table 14: **ViT-B/16** - LLaVA 1.5 (Vicuna 7B) results. We use the same suite in AM-RADIO, however we report both "Val" and "TestDev" for GQA, and also report the TextVQA score when OCR tokens are not provided as part of the context.

| Method ↓ | DFN CLIP $(\cdot 1^{-4})$ | SigLIP | DINOv2 | SAM |
|---|---|---|---|---|
| Baseline - MSE | 5.0200 | 1.9030 | 0.9591 | **5.9970** |
| Global Stdze | 4.6640 | 1.8620 | **0.6924** | 7.9080 |
| Standardize | 4.6520 | 1.8560 | 0.7036 | 7.7030 |
| **PHI-S** | **4.6310** | **1.8460** | 0.6961 | 7.7190 |

Table 15: **ViT-L/16** - Mean Squared Error for matching the teachers different algorithms. Lower values are better.

| Method ↑ | Classification | | Segmentation | | | Probe 3D | | | |
|---|---|---|---|---|---|---|---|---|---|
| | Zero Shot | kNN | ADE20k | VOC | SAM COCO | Depth | Surface Normals | Multi-View | SPair 71k |
| Baseline - MSE | 71.32 | 78.80 | 47.01 | 82.62 | **72.91** | 80.21 | 57.50 | 48.44 | 35.67 |
| Global Stdze | 78.59 | 83.15 | 50.94 | 85.58 | 71.23 | 84.51 | 60.27 | 57.86 | 52.24 |
| Standardize | 78.67 | 83.05 | **51.27** | 84.79 | 71.69 | 84.04 | 60.27 | **58.34** | 52.42 |
| PHI-S | **78.68** | **83.16** | 51.23 | 85.73 | 71.12 | **84.77** | **60.61** | 58.22 | 51.74 |

Table 16: **ViT-L/16** - Classification accuracy using both Zero Shot (DFN CLIP text encoder) and kNN. ADE20k and VOC are semantic segmentation linear probe results using 512px resolution (see Ranzinger et al. (2024) for details), and SAM COCO instance segmentation, also defined in AM-RADIO. We also show the Probe 3D El Banani et al. (2024) metrics as also reported in AM-RADIO.

| Method ↑ | GQA | | TextVQA | | POPE | VQAv2 |
|---|---|---|---|---|---|---|
| | Val | TestDev | Tokens | No Tokens | | |
| Baseline - MSE | 69.70 | 62.11 | 48.88 | 21.21 | 85.72 | 75.44 |
| Global Stdze | **71.65** | 63.08 | **53.15** | 31.43 | 86.03 | **78.37** |
| Standardize | 71.44 | **63.11** | 52.97 | 33.56 | 86.21 | 78.19 |
| PHI-S | 71.46 | 63.07 | 52.88 | **33.67** | 86.29 | 78.31 |

Table 17: **ViT-L/16** - LLaVA 1.5 (Vicuna 7B) results. We use the same suite in AM-RADIO, however we report both "Val" and "TestDev" for GQA, and also report the TextVQA score when OCR tokens are not provided as part of the context.

### A.9.4 FIDELITY

Similar to the concept of fidelity in Stanton et al. (2021) for the ability of the student to match the teacher distribution in the classification setting, it's useful to consider a similar property for generalized distribution matching. In this case, we consider how similar the student distribution is to the teacher, relative to the teacher distribution itself. Particularly,

$$F^{(k)} = \frac{\text{Var}\left[\mathbf{Y}^{(k)}\right]}{\text{MSE}\left(\mathbf{X}^{(k)}, \mathbf{Y}^{(k)}\right)} \tag{51}$$

with $\mathbf{X}^{(k)}$ being the students approximation of teacher distribution $\mathbf{Y}^{(k)}$. A value $F^{(k)} = 1$ means that the student is no better than random guessing by sampling from the the teacher's $\mathcal{N}(\mu, \sigma)$ distribution. Values $F^{(k)} > 1$ imply the student does better than random. This formulation allows us to use the geometric mean across teachers to derive a single metric of fidelity across all teachers simultaneously. We show the results for ViT-B/16 in table 18 and ViT-L/16 in table 19. Similar to previous findings, using standardization works the best, with PHI-S being best for both model sizes. When comparing standardization and whitening, the teacher distributions are always normalized to have zero mean and unit standard deviation, which means that all of the relative weightings are identical, and differences in fidelity would arise due to conditioning of the optimization.

| Method ↑ | DFN CLIP | SigLIP | DINOv2 | SAM | GeoMean |
|---|---|---|---|---|---|
| MSE | 1.1461 | 1.2077 | 1.6079 | **3.4842** | 1.6687 |
| Cosine | 0.0551 | 0.7159 | 0.9629 | 0.8118 | 0.4190 |
| Hyb MSE | 0.7783 | 1.2295 | 1.8375 | 3.0405 | 1.5206 |
| Hyb SmL1 | 0.5918 | 1.1984 | 1.9000 | 2.6184 | 1.3706 |
| Global Stdze | 1.2298 | 1.2379 | **1.9672** | 2.6706 | 1.6817 |
| Standardize | 1.2299 | 1.2362 | 1.9392 | 2.7231 | 1.6833 |
| PHI-S | **1.2355** | **1.2451** | 1.9530 | 2.7212 | **1.6909** |
| PCA-W | 1.2185 | 1.2117 | 1.8584 | 2.5972 | 1.6338 |
| ZCA | 1.2190 | 1.2120 | 1.8574 | 2.6046 | 1.6351 |
| HCA | 1.2186 | 1.2110 | 1.8565 | 2.5997 | 1.6336 |
| Ada - MSE | 1.2203 | 1.2289 | 1.8051 | 2.5915 | 1.6275 |
| Ada - PHI-S | 1.2213 | 1.2289 | 1.8063 | 2.6076 | 1.6306 |

Table 18: **ViT-B/16** - Fidelity metrics based on normalization method. Higher values are better, and higher geometric mean implies better fidelity across the set of teachers.

| Method ↑ | DFN CLIP | SigLIP | DINOv2 | SAM | GeoMean |
|---|---|---|---|---|---|
| MSE | 1.1617 | 1.2438 | 1.8051 | **3.7812** | 1.7721 |
| Global Stdze | 1.2536 | 1.2752 | 2.4606 | 2.9438 | 1.8447 |
| Standardize | 1.2503 | 1.2711 | **2.5004** | 2.8674 | 1.8373 |
| PHI-S | **1.2593** | **1.2822** | 2.4871 | 2.9377 | **1.8533** |

Table 19: **ViT-L/16** - Fidelity metrics based on normalization method. Higher values are better, and higher geometric mean implies better fidelity across the set of teachers.

### A.10 COMPARISON WITH RECENT AGGLOMERATIVE MODELS

Along with AM-RADIO at CVPR, Theia Shang et al. (2024) has been published to CoRL, and there are recent preprints for UNIC Sariyildiz et al. (2024), and UNIT Zhu et al. (2024). We report benchmarks that are common amongst the papers in table 20. For each model, we report the numbers from the original papers without attempting replication. We do run linear probing for Theia on the ADE20k task using our harness as it allows for the only common task that all papers report. We confirmed the mIoU numbers with the authors before reporting them here. We also note that the settings, such as training dataset, resolution, set of teachers, and desired outcomes, are different between the models, which means there are numerous confounding factors preventing the comparison from being "fair". We also note that the recipes for UNIC's ViT-B and ViT-L are different from each other, confounding the interpretation of the scalability of their technique. We also note the substantial strength of our produced ViT-B/16 models' semantic segmentation, where they have higher mIoU than their best teacher (DINOv2-g-reg, 48.7), despite having only 9% of the parameters. We also note that the ViT-B model is over 11% (absolute) better than the next best agglomerative model of the same size.

In table 21, we demonstrate the effectiveness of our method by including results of matching AM-RADIO's (Ranzinger et al., 2024) training recipe exactly, aside from applying the PHI-S+MSE feature distillation loss, and also using the smaller ViT-B/16 and ViT-L/16 models. Zero-shot and dense perception slightly improves with this change, while VLM metrics degrade versus SigLIP teacher (or recipe). Regardless, the choice of OpenAI CLIP versus SigLIP does not change the gen-

| Method | Model | ImageNet-1K Classification | | | Segmentation ADE20k |
|---|---|---|---|---|---|
| | | Zero Shot | kNN | Probe | |
| AM-RADIO | ViT-H/16 | **82.93** | **86.06** | - | 51.34 |
| Theia | ViT-B/16 | - | - | 75.2 | 35.61* |
| UNIC | ViT-B/16 | - | - | **83.2** | 37.3 |
| | ViT-L/14 | 81.4 | 85.6 | - | 48.3 |
| UNIT | ViT-H/14 | 78.76 | 84.18 | - | 50.19 |
| PHI-S-RADIO | ViT-B/16 | 74.57 | 82.29 | - | 48.94 |
| | ViT-L/16 | 81.01 | 84.93 | - | 51.47 |
| PHI-S OAI-CLIP | ViT-B/16 | 76.03 | 82.51 | - | 49.55 |
| | ViT-L/16 | 81.78 | 84.50 | - | **51.50** |

Table 20: Comparison between shared metrics of different agglomerative model approaches. UNIC's ViT-B/16 benchmark comes from their Table 1, and ViT-L/14 from their Table 2. UNIT's ViT-H/14 results from their Table 2. "PHI-S OAI-CLIP" uses AM-RADIO's exact training recipe and teacher set, only varying the model size.
*Our replication*

| Model | Params (M) | ImageNet1K | | Segmentation (linear) | | Vision-Language (LLaVa-1.5) | | | | SAM |
|---|---|---|---|---|---|---|---|---|---|---|
| | | Zero-shot | k-NN | ADE20k | VOC | GQA | POPE | TextVQA | VQAv2 | COCO |
| AM-RADIO (-H) | 653 | **82.93** | **86.06** | 51.34 | 84.71 | 63.01 | 86.20 | 56.32 | 79.28 | **76.23** |
| PHI-S-RADIO-B | 98 | 74.57 | 82.29 | 48.94 | 84.35 | 63.49 | 86.82 | 57.64 | 79.33 | 73.87 |
| PHI-S-RADIO-L | 320 | 81.01 | 84.93 | 51.47 | 85.49 | **64.29** | 86.86 | **62.48** | **81.10** | 75.06 |
| OAI-CLIP-B | 98 | 76.03 | 82.51 | 49.55 | 84.43 | 62.90 | 85.78 | 55.39 | 78.50 | 73.84 |
| OAI-CLIP-L | 320 | 81.78 | 84.50 | **51.50** | **85.60** | 63.47 | **87.12** | 58.46 | 79.56 | 75.32 |

Table 21: A reproduction of table 2 (first three rows), however we also show results for models trained with PHI-S using AM-RADIO's training recipe, termed "OAI-CLIP" to represent using OpenAI CLIP (Radford et al., 2021). The only differences in recipe are the feature distillation loss as studied in this work, as well as using smaller ViT-B/16 and ViT-L/16 models to maintain compute tractability.

eral conclusion that we improve upon AM-RADIO, demonstrating the effectiveness of our approach in a fair setting, even despite using smaller models.

