# OpenReview forum: "PHI-S: Distribution Balancing for Agglomerative Models"
_ICLR.cc/2025/Conference — Submitted to ICLR 2025_

### Official Review · Reviewer_JmDF · 2024-11-02

**Soundness:** 2
**Presentation:** 2
**Contribution:** 1
**Rating:** 5
**Confidence:** 2

**Summary:**

The paper is interested in improving the agglomerative model distillation, a setting in which several teachers with different heterogenous representations are used to distilled a single student.

In this context, the author identify the fact that the representations of each teacher can be widely different, having different distribution and variances. This causes challenges on the optimization side, as the losses coming from a few different teachers might have a disproportionate effect on the overall summed loss, biasing the distillation toward those few teachers (the other ones being ignored).

To alleviate this, the authors propose a new normalization technique to apply on the target representations of the teachers, PHI-S, for PCA Hadamard Isotropic Standardization, that is invariant to data rotations (contrary to standard normalization). When equipped with this normalization, the authors report that the distillation is less biased toward SAM representations (which have the largest variance) and produces more balanced results on downstream tasks than the standard MSE loss.

The authors also compare their PHI-S normalization to the other normalization schemes that could be applied on the target teacher representations.

The benchmarks include:
* Zero shot classification
* Segmentation
* VQA
* Probe 3D

**Strengths:**

* In depth analysis of the effect of different normalization schemes on representations
* Building the PHI-S normalization, and great illustration of its effect and its invariance to rotation
* A solid list of benchmarks, from various truly heterogenous teachers

**Weaknesses:**

The main weakness of the paper is the lack of real performance improvement coming from PHI-S when compared to more standard normalization schemes.

In Table 4, on ViT-L/16, where the authors mention that PHI-S is more dominant, we can see that the average rank of PHI-S places it as one of the best normalization technique. However, the ranks hide the fact that most differences are tiny and not very significant.
* When we look at Table 16, we see that rank 1 on classification is only due to 0.01% difference with standard standardization
* When we look at Table 17, we see that all normalization techniques are within less than 0.3% for most benchmarks

What those table seems to show (to me) is that in general normalization is really important, but the type of normalization used itself is not that impactful.

Overall, I think this normalization technique is interesting but the application to heterogenous teacher distillation is not that impactful.

**Questions:**

n/a

---

> ### Author Response · Authors · 2024-11-13
>
> Thank you for your review. We agree that the effect size is small on a per-benchmark basis, but rather argue that the performance across a broad spectrum of tasks does meaningfully separate different normalization techniques. Because we’re not introducing any new parameters or data, and because the student model is extremely compressed, it’s unsurprising that the effect sizes can be small, but that doesn’t make them irrelevant. Where PHI-S truly shines is in its ability to match all teachers simultaneously (figure 3, tables 3, 11, and 15) without introducing additional parameters (based on the argument that the linear projection of PHI-S can be rolled into the final adaptor linear layer, thus being representationally equivalent). We also demonstrate some interesting properties of the errors of the student model in table 5, showing how the errors are more uniformly balanced, which can be a useful property when applying euclidean measures (e.g. L2 distance) on the produced features. We also show in figure 9 a stability argument, where PHI-S clearly has the most tightly controlled error distributions, which can meaningfully impact learning dynamics.
>
> One thing that seems inarguable from the study is that normalizing the teachers is incredibly important. Once you accept that and introduce the code to support this, then there’s not really much difference in algorithmic complexity between regular standardization and PHI-S, but PHI-S makes the student model better.

---

> > ### Author Response · Authors · 2024-11-25
> >
> > Please also see our 3-part rebuttal comments regarding the use of ordinal ranks and average rank starting at https://openreview.net/forum?id=rP7rghI7yt&noteId=QqqEehzha6 as it demonstrates that PHI-S not only is superior due to rank aliasing (a concern you have raised), but continues to be superior using aggregation methods that take into account the magnitude differences in the benchmarks.

---

### Official Review · Reviewer_VJ1K · 2024-11-04

**Soundness:** 3
**Presentation:** 3
**Contribution:** 3
**Rating:** 8
**Confidence:** 5

**Summary:**

This paper presents a comprehensive study for training agglomerative models, that is to distill multiple diverse teacher models into a student model without labels by collectively "matching/aligning" the activation distributions, where the teacher models' activations are often significantly different from each other.  This paper is built on a prior art, the AM-RAIDO method and explores, from  the perspective of statistical normalization,  many different  designs of normalizing the activations of teacher models in the distillation loss functions. Based on the Hadamard matrices,  the identified PHI (PCA-Hadamard Isotropic) standarization method works the best in terms of training the best student model across different tasks. In experiments, the proposed method is tested in distilling a diverse set of foundation models (DFN CLIP, SigLIP, DINOv2 and SAM) into ViT-B/16 and ViT-L/16.

**Strengths:**

+ The proposed method is built on a solid empirical observation by accounting for the diverse distributions of different teachers' activations.
+ The paper is well written and easy to follow.
+ The proposed empirical study is comprehensive in seeking the ``best" activation alignment space.
+ The identified PHI standarization method works well in experiments compared to baseline approaches.

**Weaknesses:**

- The motivation of distilling multiple foundation models into a student model could be elaborated. Although it is an interesting problem, what is the long-term vision? Will it be practically possible to train a student model that is smaller than all teacher model, yet works comparably well in a broad sense. For example, DINOv2 can produce meaningful latent features that are useful in many downstream tasks beyond those tested in the paper.  Will the distilled student be able to retain those?
- The proposed method is trained without labels. It might be useful to discuss what effects the training datasets could have considering that different teacher models have been trained with diverse datasets.
- It might be useful to investigate the effects of batch sizes in computing the PHI-S in distillation.
- Although the propose method shows competitive performance using ViT-B/L models in comparisons to the ViT-H trained by AM-RADIO, it might be useful to train a ViT-H using the proposed method for a broader understanding of the competitiveness.
- The proposed method is computational expensive (e.g. 14080 total GPU hours for the ViT-B/16 model). It might be useful to compute the overhead of the propose PHI-S in comparisons.

**Questions:**

Overall, this is a good paper. The reviewer would like to see the authors' rebuttal on the general questions listed in the weaknesses.

---

> ### Author Response · Authors · 2024-11-13
>
> Thank you for your review. At a high level, we relied on AM-RADIO to motivate the agglomerative model paradigm in general, which allowed us to focus solely on improving it. We think that all of the issues you raised in the weaknesses section are indeed quite valid questions regarding agglomerative modeling, and we think that your first two identified weaknesses are worthy of much deeper and dedicated studies. There is emerging evidence that these models do generalize well in practice, which we talk about in our specific answer regarding your first identified weakness.
>
> **Weakness 1**: The AM-RADIO paper does most of the motivating for distilling multiple foundation models, and this work largely assumes that the technique is important based on AM-RADIO’s assertion. What they found was that different VFMs have different strengths and weaknesses, owing largely to how they were trained. The CLIP family of models is great for holistic reasoning, and seems to have a strong sense of semantic alignment with language, however they struggle significantly with dense perception tasks (e.g. semantic segmentation, or the 3D vision tasks tested here). DINOv2 is pretty much the gold standard for perception right now, but it struggles a lot with OCR (e.g. TextVQA in Table 1 of AM-RADIO). It indeed seems to be the case that fusing multiple teachers into these smaller students works well across a spectrum of tasks [6, 8, 9], and in all of these instances, the student model was smaller than even just DINOv2, ignoring the other teachers.
>
> **Weakness 2**: We agree that choice of dataset can potentially be significant. We purposefully chose to keep the dataset constant in this paper so that we could solely focus on the topic of distribution normalization. Teacher selection and dataset selection are topics that need larger treatments that couldn’t fit into the scope of this study.
>
> **Weakness 3**: Are you referring to how quickly our estimates of the rotation matrix and alpha scalar converge? We’re currently just computing these estimates over the first 3 million examples seen during training (first 3k mini-batches), but agree that it might be interesting to see how quickly those values converge. We did run into issues with PyTorch’s eigh call not returning stable eigenvectors, which was a problem for all methods relying on that function, except for ZCA, as the inverse rotation was counteracting the instability.
>
> **Weakness 4**: We agree that ultimately increasing the scale of the model would be good, but we were bound by computation constraints, as you identified in the next weakness.
>
> **Weakness 5**: Finding PHI-S is mostly negligible, as we only "learn" it over the first 3k steps (of 300k), and afterward the  application of it is a GEMM at each iteration, which is very small relative to the actual model. During inference, all of the studied normalizations can be rolled into the weights of the teacher adaptors, making them free. That said, explicitly, running the GEMM on an A100 with an input size of `(32, 729, 1280)` (from DFN CLIP) and PHI-S transform `(1280, 1280)` takes about 0.46ms, and so we’re looking at roughly 2ms/step of overhead across the four teachers, compared against the approximately 355ms/step of the general training harness. We also note that while training the model is expensive, and while specifically trying to make the training cheaper wasn’t a focus of the study, we did improve over AM-RADIO’s recipe which is 17,408 GPU-hours.
>
> **References**:
>
> [6] Drozdova, Mariia et al. “Semi-Supervised Fine-Tuning of Vision Foundation Models with Content-Style Decomposition.” ArXiv abs/2410.02069 (2024)
>
> [8] Lu, Yuxiang et al. “Swiss Army Knife: Synergizing Biases in Knowledge from Vision Foundation Models for Multi-Task Learning.” (2024).
>
> [9] Guo, Pinxue et al. “VideoSAM: Open-World Video Segmentation.” (2024).

---

> > ### Comment · Reviewer_VJ1K · 2024-11-23
> > **Thank you for your rebuttal**
> >
> > Most of my concerns have been addressed.
> >
> > It will be nice to see what workarounds could be exploited to address weakness 3, and what effects those will result in.

---

> > > ### Author Response · Authors · 2024-11-25
> > >
> > > Thinking more about your identified weakness 3, we just want to make sure that it's clear that we're not computing PHI-S on the fly for each batch. Instead, we collect the statistics for ~3 million images and compute the PHI-S estimate from that. From then on, we're just using that estimate, so it's irrespective of batch size, and the whole process simplifies to a GEMM.
> > >
> > > As far as during the estimation phase, batch size is also not a major detail, as it could be any batch size, we're just running until we have 3M samples. In the supplemental, we've provided the code that is calculating the mean and covariance matrix on the fly, so memory is not a concern.

---

### Official Review · Reviewer_QTud · 2024-11-04

**Soundness:** 2
**Presentation:** 2
**Contribution:** 2
**Rating:** 3
**Confidence:** 3

**Summary:**

This paper addresses the problem of learning a unified agglomerative model for vision - by distilling knowledge from multiple vision foundation models, used as teacher models. This work builds upon AM-RADIO (Ranzinger et al. (2024)) and identifies a possible limitation that could arise from the differences in the distributions of the features from different models, With this motivation, the paper explores different statistical normalization techniques to improve the teachers' features. Specifically, the paper introduces PCA-Hadamard Isotropic Standardization (PHI-S) that is claimed to produce the best student model compared to prior baselines and other feature standardization techniques.

**Strengths:**

- The problem of distributional differences identified by the paper with regards to the AM-RADIO work is well motivated and can be useful not just for agglomerative foundation models but also for other problems involving distillation of multiple features.
- The presentation of the different standardization techniques and the theoretical analysis of their error properties is good.

**Weaknesses:**

- The experiments initially focus on the teacher-matching metrics, for example the MSE loss to different teachers in Table 3. However, it is not clear if the a decreasing the MSE loss to a specific teacher necessarily brings downstream benefits. For example, consider the Cosine method for the DFN-CLIP teacher. Its MSE loss is 10-25 times higher than all the other methods (the Cosine method is also the worst for SigLIP). DFN-CLIP and SigLIP are the only strong performing models in zero-shot Imagenet classification (see Table 1 in [1]). One would expect higher MSE for DFN-CLIP and SigLIP to result in poorer performance in zero-shot Imagenet classification. But the finding is to the contrary - the Cosine method performs best in zero-shot Imagenet classification among non Ada- methods (see Table 13). If the initial MSE loss is mainly resulting from the difference in feature norms, then the experiments that depend primarily on the MSE loss do not make sense.
- The empricial comparison of PHI-S to AM-RADIO in Table 2 is not a fair comparison as the teacher models used are different between the 2 works. So, this comparison is not a fair demonstration of the benefits obtained by using the feature standardization proposed in this work.
- The empirical performance on downstream tasks are presented in the main paper based on average ranks on different groups of tasks. This makes it difficult to get a clear understanding of the performance and does not provide a clear comparison between the different methods. Looking at the performance metrics in Tables 13-17, the performance of PHI-S is only on par with other baseline methods or the improvement is too marginal to be considered as a significant improvement. The presentation in terms of ranks is also somwhat misleading when the performance difference between the ranks are so small/insignificant. For instance, one could argue that adding the AdaLoss to MSE brings consistent and significant improvements across all tasks (Ada-MSE vs MSE) but the performance difference between the other methods are comparatively insignificant. This raises the question of whether the usage of Hadamard matrices for standardization (which is claimed to be one of the contributions of the paper) is actually effective.
- The paper is heavily reliant on AM-RADIO [1]. The proposed standardization solution is not completely new [3, 4] but it can still be interesting to show that it can be useful in new applications. However, the effectiveness of the proposed solution is not sufficiently demonstrated in this paper.
- DINOv2 also brings additional benefits like transfer learning, domain generalization and robustness properties. The paper lacks analysis of other such beneficial properties of the teacher models. This can be a worthy addition that is not explored in [1] either.

- Minor: Inconsistent notations can be avoided - in lines 127-133, the different teachers are indexed by $(t)$ whereas in section 2.2 they are indexed by $k$.
- Minor: The readability of the paper can be improved by formatting the references appropriately in parentheses when they are not part of the sentence.

[1] Ranzinger, Mike, et al. "AM-RADIO: Agglomerative Vision Foundation Model Reduce All Domains Into One." CVPR 2024.

[2] Hu, Hanzhang, et al. "Learning anytime predictions in neural networks via adaptive loss balancing." Proceedings of the AAAI Conference on Artificial Intelligence 2019.

[3] Kanj, Hind, et al. "A comparative study of the whitening methods in linear video coding and transmission schemes." 2022 11th International Symposium on Signal, Image, Video and Communications (ISIVC). IEEE, 2022.

[4] Pratt, William K., Julius Kane, and Harry C. Andrews. "Hadamard transform image coding." Proceedings of the IEEE 57.1 (1969).

**Questions:**

- Adding the AdaLoss to MSE significantly improves over MSE results in Tables 13 and 14 on ALL tasks. Have you experimented with adding the AdaLoss to some other methods that perform well on dense tasks such as Hyb SmL1 or Standardize?
- Different teacher models also normalize the features in different ways during the pre-training step. For example, the features in the DINO head are L2 normalized and lie on a unit hypersphere. Some models use LayerNorms on the output features. Could there be a benefit in taking into account the normalizations used during pre-training while selecting appropriate feature standardization?
- What is the correlation between the teacher-matching loss and the downstream performance? For example, if the loss to a specific teacher is minimized, should that lead to the resultant model displaying properties most similar to that teacher? If the goal is to demonstrate the effectiveness of the feature standardization and its effect on downstream tasks, would it make more sense to consider teacher models which are distinctly good at specific tasks but significantly worse at other tasks? This might enable one to more clearly evaluate the downstream impact of different teachers and minimizing their losses.

Update after rebuttal (2024-11-24 AOE): I have reviewed the rebuttal from the authors and since my earlier concerns are mostly unaddressed, I would like to keep my score.

---

> ### Author Response · Authors · 2024-11-13
>
> Thank you for your in-depth review. We agree that the correlation between fidelity and downstream benchmarks is not 1-to-1, which can be seen in how the relative differences on particular benchmarks is not straightforward. We opted to rely on summarizing the entire suite of benchmarks, under the assumption that a model that tends to perform better across the suite would tend to be a more generalizable model. Rank works well in this regard because we don’t have to normalize the benchmark units in order to aggregate a different way. The primary reason we have the “Avg No COCO” and “Avg No MSE/COCO” columns in Table 1 is so that we could demonstrate that PHI-S works without relying on outlier scores in the MSE task. However, we didn’t have a column which includes everything _except_ for MSE, which we’ve rectified in the updated draft. The added column “Avg No MSE” in table 1 has a consistent conclusion as before, which is that PHI-S does the best on average, followed by global standardization. We did the same thing for table 4, which still has PHI-S as the best method, and in this case, regular standardization remains in second place. We have a detailed response below.
>
> **Weakness 1**: We’re applying distribution balancing to the spatial features of the teacher models, whereas zero shot accuracy is affected by the summary loss (always cosine) between the student and teacher. The CLIP teachers have a supervised summary feature, and it’s trained using cosine similarity with their text encoders. DINOv2 also uses cosine similarity for its summary tokens. Presumably this is why AM-RADIO chose to use cosine similarity loss for the summary of all teachers (SAM withstanding). There is a tension between the summary losses (cosine) and the feature losses (studied in our paper), where changing the weight of one will impact the others.
>
> **Weakness 2**: There are two different statements that the paper is making. The central table for the purposes of the paper is actually table 1, where the setup is tightly controlled between different feature distillation losses, and thus it’s a fair comparison. Table 2 is instead showing that we were able to apply the techniques studied to produce an agglomerative model that is competitive with AM-RADIO, but at 15% and 49% of the parameters for ViT-B and ViT-L respectively. It’s essentially a hook for “why should I care” as opposed to the central result of the study.
>
> **Weakness 3**: We used the comparative ranks as it’s not clear which benchmarks are most important, and also how their magnitude changes relative to each other is important. Instead, the average rank argument is that across a large swath of benchmarks, a better model will tend to do better than its peers across a wider range of tasks. A major problem with agglomerative modeling (or even foundation models in general) is that the downstream use case is unclear, and so the choice of which metrics to target is subjective. If we substantially changed the weights across all summary and feature losses in favor of say, DFN CLIP’s summary loss, then we’d expect to get very strong zero shot (and probably kNN) accuracy, but it would come at the expense of the dense tasks. Applying AdaLoss to the MSE baseline actually does have an effect similar to the standardization methods, as it’s applying the inverse expected loss. This is why we also studied AdaLoss + PHI-S, and found improved teacher matching metrics (table 11) across the board, suggesting that its inclusion, all things equal, yields students with higher fidelity. All said, however, AdaLoss tended to work worse across the full task suite as compared to the standardization methods (Global, Standardize, PHI-S) aside from classification where it strongly produces the best results (see for example the Probe 3D metrics in table 13).
>
> **Weakness 4**: We disagree on the claims of limited novelty. As for the comparison against AM-RADIO, the entire study focuses around a single design decision in their paper, their equation 3. So we’re leveraging their work, not necessarily comparing against it, aside from in table 2 where we eventually bring the details full circle. While the applications of Hadamard matrices are not new, to the best of our knowledge, combining the eigenvector projection of PCA with the Hadamard rotation as a method to get identical variance across the dimensions of a multivariate distribution is entirely novel. Even Hadamard whitening (as described in [3], but also [5]) is not the same as we’re employing it here. In [3, 5] they’re spreading the error in a similar motivation as ours, but not applying statistical whitening (multiplying by the inverse root eigenvalues). They’re especially not using the Hadamard matrix for statistical standardization, nor are they motivated by the isotropic property of our approach.

---

> > ### Comment · Reviewer_QTud · 2024-11-25
> > **Response to authors**
> >
> > Thanks for the detailed rebuttal response. I include my further responses to each point below:
> >
> > **Weakness 1** - The main contribution of this work revolves around standardization of representations from the multiple teachers. This is expected to better balance the information learned from the representations of the different teachers. If there is little correlation between fidelity and downstream benchmarks, the motivation for this work remains weak. This concern still remains unaddressed.
> >
> > **Weakness 2** - The concern raised here mainly relates to the usage of different teachers compared to AM-RADIO (one of the teachers is changed from OpenAI [OAI] CLIP to SigLIP). So, compared to AM-RADIO, it is unclear if the claimed superior performance comes from changing this teacher or from the specific choice of standardization (main contribution of this paper). I note that the authors have added an additional comparison in Table 21 that shows the performance of PHI-S with OAI CLIP teacher. Here, we clearly observe that using the SigLIP teacher instead of the OAI-CLIP brings significant performance improvement on the LLaVa-1.5 tasks. These tasks also happen to be the ones where PHI-S has a larger margin of improvement over AM-RADIO. If we consider Hyb-SmL1 loss as the closest one to AM-RADIO, the performance gap between Hyb-SmL1 and PHI-S is small on all except 2 tasks (see Table 13 and 14).
> >
> > **Weakness 3** - The usage of ranks to simplify the presentation of results summarizing multiple downstream tasks is reasonable when the underlying metrics are still significant. However, in this case, the actual performance metrics shown in the appendix tables are so close between the different methods. So, basing conclusions entirely on average ranks seems to hide the fact that the performance difference between certain ranks are insignificant. The importance of the fidelity experiments is unclear as it poorly correlates to the real target, that is downstream performance.
> >
> > **Weakness 4** - I agree with the authors regarding the novelty in the proposed PHI-S standardization method. The method does improve fidelity which measures how well the agglomerative model is able to predict the representations of the different teachers. But this correlates poorly with the downstream metrics, as acknowledged by the authors as well. In addition, due to the concerns explained above in weakness 2 (problems related to usage of average ranks), the effectiveness of PHI-S in improving downstream performance is not demonstrated clearly.
> >
> > **Weakness 5** - I agree that teacher selection is well out-of-scope for this study. However, it can be valuable for this study to include a discussion on what properties of the teacher models are retained in the agglomerative model (I used DINOv2 to merely provide a few examples of properties which could be studied). This is especially relevant to this work which studies how to balance the contribution of different teachers.
> >
> > **Question 3** - In the example with two teachers being distilled into a student, let the model be able to perfectly match the two teachers. The student representations could still be highly entangled, meaning that some properties like linear separability of classes in one of the teacher representations is not guranteed in this student. This makes it questionable if higher fidelity of the representations produced by the adapter heads specific to each teacher is actually important for learning good student representations.
> >
> > As exaplined above, most of the concerns raised in the weaknesses remain unaddressed. Hence, I would like to keep my score.

---

> > > ### Author Response · Authors · 2024-11-25
> > > **Rebuttal #2 (Part 1)**
> > >
> > > Thank you again for your attention to detail. It appears as though the major issue with the methodology that you've identified is with the choice to use ranks, and average rank, to draw conclusions about the method suitability. The key argument in favor of your point is that rank erases the magnitude of difference between two methods, thus allowing a method that was only marginally better than another on all but one task, and meaningfully worse on the final task to still win due to the small wins elsewhere. Hopefully you agree with this characterization, but please clarify if this interpretation is wrong.
> > >
> > > Partly, we were hoping rank would work as a shorthand for more complicated ways to parse the data, but we also present other methods of aggregating this data, and all still show that PHI-S was the dominant technique studied. In the following, we are going to completely ignore fidelity, and instead focus solely on the benchmarks, and using the data from tables 13 and 14.
> > >
> > > ## Simple Average
> > > ### Raw Average Across Metrics
> > >
> > > In this case, we simply took the average of all benchmark metrics, without accounting for differences in units, and also without accounting for strong benchmark correlations (e.g. GQA-Val and GQA-TestDev are treated independently)
> > >
> > > | Method                 | Average |
> > > |------------------------|---------|
> > > | Baseline - MSE         | 58.72   |
> > > | Baseline - Cosine      | 63.36   |
> > > | Baseline - Hybrid MSE  | 63.31   |
> > > | Baseline - Hybrid SML1 | 63.49   |
> > > | Global Stdze           | 63.61   |
> > > | Standardize            | *63.76*   |
> > > | PHI-S                  | **63.80**   |
> > > | PCA-W                  | 63.65   |
> > > | ZCA                    | 63.67   |
> > > | Hadamard               | 63.61   |
> > > | AdaLoss - Baseline     | 63.29   |
> > > | AdaLoss - PHI-S        | 63.17   |
> > >
> > > ### Raw Average With Aggregate Benchmarks
> > >
> > > To deal with heavily correlated benchmarks, we averaged across the different tasks and subtasks, as detailed in section 4 of the paper.
> > >
> > > | Method                 | Classification | Semantic Segmentation | SAM COCO | LLaVa 1.5 | Probe 3D | Average |
> > > |------------------------|----------------|-----------------------|----------|-----------|----------|---------|
> > > | Baseline - MSE         | 63.86          | 60.25                 | **71.90**    | 63.00     | 53.51    | 62.50   |
> > > | Baseline - Cosine      | 75.59          | *65.70*                 | 69.42    | 66.01     | 57.84    | 66.91   |
> > > | Baseline - Hybrid MSE  | 74.03          | 65.65                 | *70.54*    | 65.97     | 58.30    | 66.90   |
> > > | Baseline - Hybrid SML1 | 75.34          | 65.53                 | 69.53    | 66.29     | 58.18    | 66.97   |
> > > | Global Stdze           | 75.21          | 65.48                 | 69.75    | 66.18     | 58.92    | 67.11   |
> > > | Standardize            | 74.93          | 65.33                 | 70.22    | 66.37     | 59.21    | *67.21*   |
> > > | PHI-S                  | 75.13          | **65.86**                 | 69.89    | 66.24     | *59.27*    | **67.28**   |
> > > | PCA                    | 74.77          | 65.27                 | 69.55    | 66.15     | **59.31**    | 67.01   |
> > > | ZCA                    | 74.83          | 65.32                 | 69.37    | 66.34     | 59.13    | 67.00   |
> > > | Hadamard               | 74.90          | 65.42                 | 69.19    | 66.23     | 59.04    | 66.95   |
> > > | AdaLoss - Baseline     | *76.37*          | 64.89                 | 69.60    | **66.48**     | 56.65    | 66.80   |
> > > | AdaLoss - PHI-S        | **76.38**          | 65.07                 | 69.74    | *66.44*     | 56.24    | 66.77   |
> > >
> > >
> > > Next, we will show these scores after normalizing them to deal with the different units in the benchmarks.

---

> > > ### Author Response · Authors · 2024-11-25
> > > **Rebuttal #2 (Part 2)**
> > >
> > > In the previous comment, we showed how PHI-S is the method of choice when using simple averages across the benchmarks, with and without task averaging. Now, we'll change the way we aggregate across tasks, dealing with the fact that each task has potentially different units and especially different magnitudes. The idea behind this is that we subtract the mean score across the methods studied for a given benchmark, and then divide by the standard deviation across the methods for the same benchmark.
> > >
> > > ## Benchmark Statistics
> > >
> > > | Method   | Zero Shot | kNN    | ADE20k | VOC    | SAM COCO | GQA Val | GQA TestDev | Text VQA Tokens | TextVQA No Tokens | POPE  | VQAv2  | Probe 3D Depth | Probe 3D Surface Normals | Probe 3D Multiview Correspondence | Probe 3D SPair 71k |
> > > |----------|-----------|--------|--------|--------|----------|---------|-------------|-----------------|-------------------|-------|--------|----------------|--------------------------|-----------------------------------|--------------------|
> > > | Mean     | 69.748    | 78.805 | 47.411 | 82.546 | 69.892   | 69.741  | 61.918      | 50.08           | 23.554            | 85.47 | 75.776 | 81.253         | 56.447                   | 53.098                            | 41.066             |
> > > | Std.Dev. | 4.393     | 2.311  | 1.619  | 1.421  | 0.733    | 0.765   | 0.796       | 0.917           | 2.887             | 0.358 | 1.128  | 1.222          | 0.633                    | 2.03                              | 3.706              |
> > >
> > > ## Normalized Scores
> > >
> > > | Method | Zero Shot | kNN | ADE20k | VOC | SAM COCO | GQA Val | GQA TestDev | Text VQA Tokens | TextVQA No Tokens | POPE | VQAv2 | Probe 3D Depth | Probe 3D Surface Normals | Probe 3D Multiview Correspondence | Probe 3D SPair 71k | Average |
> > > |---|---|---|---|---|---|---|---|---|---|---|---|---|---|---|---|---|
> > > | Baseline - MSE | -3.091 | -3.143 | -3.095 | -3.129 | **2.735** | -3.126 | -3.025 | -3.022 | -2.942 | -0.868 | -3.16 | -2.917 | -2.198 | -2.653 | -2.025 | -2.377 |
> > > | Baseline - Cosine | 0.385 | 0.403 | 0.37 | **0.594** | -0.643 | 0.365 | -0.122 | 0.175 | 0.199 | -1.92 | 0.323 | 0.423 | 0.02 | 0.211 | -0.399 | 0.026 |
> > > | Baseline - Hybrid MSE | -0.093 | -0.035 | 0.364 | *0.524* | *0.886* | 0.156 | 0.053 | 0.076 | -0.008 | -0.781 | 0.145 | -0.306 | -0.233 | -0.259 | 0.641 | 0.075 |
> > > | Baseline - Hybrid SML1 | 0.328 | 0.296 | *0.506* | 0.193 | -0.495 | 0.378 | 0.543 | 0.12 | 0.12 | 0.74 | 0.349 | **0.726** | -0.027 | 0.293 | -0.167 | 0.26 |
> > > | Global Stdze | 0.264 | 0.304 | 0.296 | 0.369 | -0.197 | *0.47* | 0.455 | 0.251 | -0.348 | *1.143* | *0.385* | *0.628* | 0.905 | 0.508 | 0.394 | *0.388* |
> > > | Standardize | 0.173 | 0.236 | 0.284 | 0.172 | 0.45 | 0.391 | 0.305 | 0.218 | 0.224 | **1.314** | 0.376 | -0.666 | 0.052 | **0.765** | **1.134** | 0.362 |
> > > | PHI-S | 0.223 | 0.312 | **0.753** | 0.383 | -0.002 | **0.6** | **0.795** | 0.185 | -0.095 | 0.133 | **0.465** | 0.521 | 0.542 | *0.687* | 0.77 | **0.418** |
> > > | PCA | 0.111 | 0.214 | 0.104 | 0.292 | -0.467 | 0.143 | 0.116 | 0.436 | 0.203 | -0.113 | 0.137 | 0.513 | 0.415 | 0.651 | *0.856* | 0.241 |
> > > | ZCA | 0.143 | 0.204 | 0.259 | 0.179 | -0.709 | 0.313 | *0.569* | 0.033 | 0.373 | 0.911 | 0.216 | 0.145 | **1.237** | 0.686 | 0.616 | 0.345 |
> > > | Hadamard | 0.164 | 0.227 | 0.265 | 0.313 | -0.957 | 0.273 | -0.16 | -0.175 | 0.428 | 0.393 | 0.261 | 0.292 | *0.985* | 0.617 | 0.559 | 0.232 |
> > > | AdaLoss - Baseline | **0.716** | *0.453* | -0.106 | -0.011 | -0.398 | 0.012 | 0.493 | *0.807* | **1.065** | -1.148 | 0.278 | 0.259 | -0.185 | -0.61 | -1.138 | 0.033 |
> > > | AdaLoss - PHI-S | *0.678* | **0.529** | -0.001 | 0.123 | -0.203 | 0.025 | -0.022 | **0.895** | *0.781* | 0.195 | 0.225 | 0.382 | -1.513 | -0.896 | -1.243 | -0.003 |
> > >
> > > Again we have the issue of correlated benchmark tasks, so in the next comment, we'll perform the same common-benchmark-aggregation with these normalized scores.

---

> > > ### Author Response · Authors · 2024-11-25
> > > **Rebuttal #2 (Part 3)**
> > >
> > > Finally, we show the aggregated benchmark scores after normalization:
> > >
> > > ## Normalized Average with Aggregate Benchmarks
> > >
> > > | Method | Classification | Semantic Segmentation | SAM COCO | LLaVa 1.5 | Probe 3D | Average |
> > > |---|---|---|---|---|---|---|
> > > | Baseline - MSE | -3.117 | -3.112 | **2.735** | -2.69 | -2.448 | -1.727 |
> > > | Baseline - Cosine | 0.394 | *0.482* | -0.643 | -0.163 | 0.064 | 0.027 |
> > > | Baseline - Hybrid MSE | -0.064 | 0.444 | *0.886* | -0.06 | -0.039 | 0.233 |
> > > | Baseline - Hybrid SML1 | 0.312 | 0.349 | -0.495 | 0.375 | 0.206 | 0.15 |
> > > | Global Stdze | 0.284 | 0.332 | -0.197 | 0.393 | 0.609 | 0.284 |
> > > | Standardize | 0.205 | 0.228 | 0.45 | **0.471** | 0.321 | *0.335* |
> > > | PHI-S | 0.267 | **0.568** | -0.002 | 0.347 | *0.63* | **0.362** |
> > > | PCA | 0.162 | 0.198 | -0.467 | 0.154 | 0.609 | 0.131 |
> > > | ZCA | 0.174 | 0.219 | -0.709 | *0.402* | **0.671** | 0.151 |
> > > | Hadamard | 0.195 | 0.289 | -0.957 | 0.17 | 0.613 | 0.062 |
> > > | AdaLoss - Baseline | *0.585* | -0.058 | -0.398 | 0.251 | -0.418 | -0.008 |
> > > | AdaLoss - PHI-S | **0.604** | 0.061 | -0.203 | 0.35 | -0.817 | -0.001 |
> > >
> > > Across all of the previous tables, PHI-S produced the best metrics on average, as well as by using ranks as in the paper. We strongly urge the reviewer to reconsider their position on the method being insignificant in light of this data, and we would also be willing to re-work table 1 in the camera ready to instead show the results in this table, as it then doesn't mask the differences in magnitude between studied balancing methods.
> > >
> > > Based on:
> > > * The overall study design having scientific validity
> > > * The proposed PHI-S demonstrating increased effectiveness in the domain studied
> > > * PHI-S being a general mathematical technique that broadly works on arbitrary normal multivariate distributions anywhere regular statistical standardization or whitening would be applied
> > > * The final produced models being highly competitive (and mostly better) than the current state of the art (AM-RADIO, additional reference for claim [8]), but at 15% and 49% of the size
> > >
> > > We think that this study and paper are valuable scientific contributions to the community.
> > >
> > > [8] Swiss Army Knife: Synergizing Biases in Knowledge from Vision Foundation Models for Multi-Task Learning (https://arxiv.org/abs/2410.14633)

---

> > > ### Author Response · Authors · 2024-11-29
> > >
> > > Dear reviewer QTud,
> > >
> > > Have you had a chance to review our latest remarks and tables? The tables directly address your continued concerns in weaknesses 2-4. For weakness 1, while fidelity is weakly correlated with downstream metrics on a per-benchmark basis, we will show the geometric mean fidelity from Table 18, combined with the results in table https://openreview.net/forum?id=rP7rghI7yt&noteId=J4cjteC2Rv, and then show the Spearman rank correlation between fidelity and the normalized benchmark score.
> > >
> > > | Method | GeoMean Fidelity | Avg Norm Benchmark | Fidelity Rank | Benchmark Rank
> > > |---|---|---|---|---|
> > > | Baseline - MSE | 1.6687 | -1.727 | 4 | 12 |
> > > | Baseline - Cosine | 0.4190 | 0.027 | 12 | 9 |
> > > | Baseline - Hyb MSE | 1.5206 | 0.233 | 10 | 4 |
> > > | Baseline - Hyb SmL1 | 1.3706 | 0.150 | 11 | 6 |
> > > | Global Stdze | 1.6817 | 0.284 | 3 | 3 |
> > > | Standardize | 1.6833 | 0.335 | 2 | 2 |
> > > | PHI-S | 1.6909 | 0.362 | 1 | 1 |
> > > | PCA-W | 1.6338 | 0.131 | 6 | 7 |
> > > | ZCA | 1.6351 | 0.151 | 5 | 5 |
> > > | HCA | 1.6336 | 0.062 | 7 | 8 |
> > > | AdaLoss - MSE | 1.6275 | -0.008 | 9 | 11 |
> > > | AdaLoss - PHI-S | 1.6306 |  -0.001 | 8 | 10 |
> > >
> > > Spearman rank correlation: 0.50
> > >
> > > As such, there is actually moderate correlation between fidelity and the methods studied in this work. Additionally, fidelity correctly ranked the top 3 methods based on average normalized benchmark scores. This, along with the numerous ways we've demonstrated the effectiveness of PHI-S, should assuage weaknesses 1-4.
> > >
> > > Weakness 5 is still out of scope for this study. This study is solely focused on balancing a given set of distributions, and does not seek to make value judgments on the importance of a given teacher. While these may be two sides of the same coin, PHI-S is focused on getting all of the distributions onto a level playing field, and then a subsequent work might be focused on maximizing the utility of the agglomerated model based on balancing a selected set of teachers based on importance. Agglomerative models are a relatively nascent domain, and as such, weakness 5 should be regarded as a topic worthy of future study (of which the authors wholeheartedly agree), but it's unfair to consider this is a weakness of the paper under review.
> > >
> > > We agree with your remarks in question 3. Particularly given the forced compression of our student, as it's too small to memorize all teachers, the representations are indeed entangled. However, we've now demonstrated multiple different ways that fidelity is at least a moderate predictor of downstream benchmark performance. Looking forward, having a clearer picture of what these entanglements are is a very interesting topic of future study.
> > >
> > > We request that you review these remarks, and update your score accordingly.
> > >
> > > Regards, Authors

---

> ### Author Response · Authors · 2024-11-13
>
> **Weakness 5**: We agree that DINOv2 is a hugely valuable model. Studying the benefits of the teacher models, or even teacher selection itself, is out of scope for this study. There is some recent evidence [6] that suggests that AM-RADIO enjoys better generalization and transfer learning than DINOv2. Theia [7] does study teacher inclusion, however it’s also a relatively small scale study, and they’re not trying to build a general foundation model, but a limited robotic one as evidenced by their exclusion of the summary tokens.
>
> **Weakness 6,7**: Thank you for the formatting feedback. We will incorporate it into the draft.
>
> **Question 1**: We explored combining AdaLoss with PHI-S (tables 11-14), and found that teacher MSE is reduced further when additionally applying PHI-S. We agree that AdaLoss marks a large improvement over the baseline MSE without balancing, but disagree that it’s generally competitive against naive weighting combined with standardization.
>
> **Question 2**: It’s actually the projection head of DINOv2 that is L2 normalized, but the backbone features themselves are unconstrained, and it’s the latter that we’re trying to match. SAM has supervised features at the output, but we’re feature matching before the neck, where they’re also unconstrained. DFN CLIP and SigLIP don’t apply any supervisory signal to the features. We think it would be easy to construct an example where it does make sense to take the final operator of the teacher into account before matching. This is exactly why the summary loss is cosine across the board, because the teacher models used cosine for the summary tokens, and thus it’s possible (or even likely) that the vector magnitudes are meaningless. It seems entirely plausible that LayerNorm could benefit from a boutique normalization technique that operates better for it. What we found in figure 2 was that all of the teachers had mostly gaussian distributions with various means and variances across the dimensions, which is something PHI-S is well designed to handle.
>
> **Question 3**: This is currently the major black box of agglomerative models. If the adaptor head of the student perfectly matched a teacher, then the student would have the same behaviors as the teacher, but only when still applying that adaptor. Once you introduce a second teacher, even assuming that it also perfectly matched that teacher, it’s unclear what properties the student backbone would exhibit, assuming that it wasn’t wide enough to represent both teachers exactly. In this case of our study, our ViT-B only has a width of 768, and ViT-L a width of 1024. DINOv2-g alone has a width of 1536, meaning that our student (in the case of ViT-B) is only half the width of just one of its teachers. What we do consider to be important, all things equal, is that higher teacher fidelity (measured here as lower MSE) is preferable. Our study doesn’t introduce a change of data, and we constrain the experimental setting to be identical everywhere except for the loss function (and associated weights), which makes it fair to care about fidelity a lot (tables 3, 11, and 15). We have also added section A.9.4 with associated tables 18 and 19 to attempt to unify fidelity across the teachers in a way in which it can be aggregated. We agree that finding teachers that are distinctly good at certain tasks and correlating them with their losses is a good topic of future work, but teacher selection was largely outside the scope of this study.
>
> **References**:
>
> [5] Anthony Trioux, François-Xavier Coudoux, Patrick Corlay, M Gharbi. A comparative preprocessing study for softcast video transmission. 2018 9th International Symposium on Signal, Image, Video and Communications (ISIVC), Nov 2018, Rabat, Morocco. pp.54-59, ff10.1109/ISIVC.2018.8709171ff. ffhal-03335982f
>
> [6] Drozdova, Mariia et al. “Semi-Supervised Fine-Tuning of Vision Foundation Models with Content-Style Decomposition.” ArXiv abs/2410.02069 (2024)
>
> [7] Shang, Jinghuan et al. “Theia: Distilling Diverse Vision Foundation Models for Robot Learning.” ArXiv abs/2407.20179 (2024)

---

> ### Author Response · Authors · 2024-11-20
>
> In response to weakness (2), we have also updated the paper in section A.10 and associated Table 21, which is an exact reproduction of AM-RADIO's training recipe, aside from using the smaller ViT-B and ViT-L models, and also applying PHI-S+MSE loss. This allows for a direct comparison to their results. Zero shot and dense perception metrics improve slightly over our SigLIP model with this change, while VLM metrics degrade. However, the ViT-L model is still generally better than AM-RADIO's ViT-H at half the parameter count. This demonstrates that the positive changes in quality are not attributable to the change in recipe (e.g. SigLIP or stages), but rather the specific topic studied in the paper, which is distribution normalization across the teachers.

---

### Official Review · Reviewer_Kdwv · 2024-11-04

**Soundness:** 3
**Presentation:** 2
**Contribution:** 2
**Rating:** 5
**Confidence:** 4

**Summary:**

The paper proposes a modification to the AM-RADIO multi-task distillation framework (Ranzinger et al.) to improve its performance on downstream tasks. More specifically, it focuses on the problem of normalizing the output distributions of different tasks/teachers before distilling them into a single student. The paper performs ablations on different normalization techniques such as simple mean/std global normalization, whitening, Hadamard whitening and then proposes a modified normalization technique called PHI-S which is rotation invariant. The experiments suggest that PHI-S, on average, is a better normalization for AM-RADIO compared to the other normalizations considered in the experiments.

**Strengths:**

+ The proposed PHI-S normalization is easy to be applied to the existing AM-RADIO framework.
+ PHI-S on average performs better than the other normalization techniques considered in the paper.
+ I would also like to appreciate the inclusion of recent similar techniques (such as UNIT and UNIC) in the related works.

**Weaknesses:**

**Contribution:**

The paper proposes a new normalization technique to be applied to teacher distributions in multi-teacher distillation settings. Although the importance of normalization has been previously studied in the literature and sufficiently covered in the related works by the authors, the proposed method targets a specific framework, namely AM-RADIO. To me the main finding of the paper is the importance of such normalization for this framework and introduction of the PHI-S normalization technique. However, normalization for AM-RADIO has been also recently explored (such as Theia as also acknowledged by the authors). This makes the paper an experimental follow up by introducing a new type of normalization for which I expected to see stronger experimental results (see below).

**Experimental results**:

 The main contribution of the paper is the introduction of the PHI-S normalization with the goal of improving the existing AM-RADIO pipeline. Most experiments in the paper focus on ablating and comparing PHI-S with simpler normalization techniques. Particularly, I found it hard to find a fair comparison between AM-Radio with and without PHI-S and with other existing related works. Table 2 and Table 18 try to provide such a comparison, but the settings are not apple to apple (i.e. using different models, different training, image resolutions, etc for different methods). This makes it hard to verify the main claim of the submission. Please see the questions sections for additional questions/comments.

**Paper organization/presentation:**

I suggest the authors consider re-organizing the paper. A significant portion of the main paper (page 3, page 4, and page 5) goes over the details of the previously known methods and normalization techniques . This can be greatly summarized and/or moved to the supplementary, freeing up some space for discussing the main contributions of the paper and including important experimental results.

**Questions:**

1) Table 2 suggests that the performance of the AM-RADIO zero-shot/few-shot image classification reduces when PHI-S normalization is applied. However, I see that the baseline and the proposed approach are using different models. Is the reduction in performance caused by using a different backbone or it is caused by the proposed PHI-S normalization. Does your method still hurt AM-RADIO if you switch to ViT-H? Adding apple-to-apple comparisons between the proposed method, AM-RADIO and previous approaches can help understanding the effectiveness of the proposed normalization.

2) Table 2 reports PHI-S-RADIO-L to have an ImageNet-1K accuracy of 81.01 and 84.68 on zero-shot and kNN respectively. However, Table 18 reports 80.45 and 84.57 for the same model (PHI-S-RADIO-L) and the same dataset to my understanding. What is the difference between these two experiments?

3) According to the reported experiments in the paper, the proposed PHI-S normalization almost always hurts the performance on the SAM COCO instance segmentation, even compared to the simple MSE baseline without normalization. Is there a specific reason for this observation? An analysis can help the reader to better understand the shortcomings of the proposed normalization.

---

> ### Author Response · Authors · 2024-11-13
>
> Thank you for your detailed review, particularly catching the discrepancy in your second question. At a high level, we feel as though PHI-S is applicable to any situation where you’re learning one or more fixed multivariate distributions. AM-RADIO (or the other agglomerative approaches) fall into this category, and we chose to go with AM-RADIO because it’s been around the longest. Learning a fixed distribution could also be applied in settings such as quantization, and we believe PHI-S has a useful grounding here as it would minimize quantization errors across dimensions since the distribution is balanced across. This is actually where the previous literature on Hadamard matrices in our related work section focuses, although they don’t seem to combine it with PCA, which means they aren’t identically spreading the variance. Getting an apples-to-apples comparison across the agglomerative papers is hard because the space hasn’t standardized on a single setting in the same way that DataComp did for CLIP models. Instead, we wanted our study to be internally consistent, which is Table 1, and then we wanted to demonstrate the relevance in Table 2 by showing that we can ultimately get a very strong ViT-L/16 model. We agree that having an apple-to-apples with at least AM-RADIO would be nice, however, their ViT-H is quite costly to train, and it was prohibitive for the scope of this work. In the following we go through the details of your feedback:
>
> **Contribution Weakness**: While we agree that Theia also studied standardization, PHI-S has applications beyond just what we studied in this paper. It could similarly be applied anywhere you’re trying to learn a fixed distribution, which we’re seeing similar techniques recently with quantization [10,11, 12], albeit they rely on random matrices as an approximation to PCA. Our paper provides an exact solution which is suitable in situations where the distribution is not changing. We also argue that while Theia applies standardization, they don’t deeply analyze the choice, and this study serves to reinforce the application from both a theory and applications standpoint.
>
> **Experimental Weakness**: While we do include table 2 as a means for grounding our results with the general literature, it wasn’t our goal to exactly replicate AM-RADIO, but rather rely on the fundamentals introduced in table 1 as a means for comparing the different techniques in isolation of a bunch of design choices. AM-RADIO’s ViT-H model is extremely expensive to train, and thus attempting to exactly match it was beyond our computational budget at the time, which is why we made choices such as multi-stage training for table 2 simply to reduce the computational burden. The net result is that we get rather strong results, which table 2 showcases, but really rely on table 1 as the crux of the study where fair comparisons are made.
>
> **Question 1**: Because PHI-S is a rotation and uniform scale, one way to think about what it’s doing is implicitly applying loss weights to the different teacher’s features, and a result of this re-weighting is that the summary losses are impacted. Roughly, this would be similar to loss weighting these components by $\alpha^2$ from table 8. Similar to our answer below about SAM metrics, it’s not necessarily surprising that changing the weighting between losses will affect benchmarks that are quite sensitive to particular models, in this case, the summary loss for DFN CLIP/SigLIP. The argument we’re trying to make with this paper is that it’s hard to even address high-level balancing across teachers without first getting them onto a level playing field, and we’re arguing that PHI-S does the best job across those methods studied of balancing their distributions.
>
> **Question 2**: These differences were a result of transcription errors. We’ve corrected both tables, and greatly appreciate your catching of the mistake.
>
> **Question 3**: This has to do with how dominant SAM’s feature distribution is relative to the other models. In effect, when applying simple MSE, most of the loss weight is applied to SAM, which you can see in figure 3. So the baseline MSE model is learning SAM much better, at the expense of the other models. Tables 13 and 16 show that basically any type of balancing (as they’ll all reduce the weight of SAM) has a negative effect on the SAM COCO benchmark. Because this benchmark relies on exactly (or at least acceptably) approximating the real SAM model, it’s expected that having a higher SAM MSE (figure 3) would lead to reduced SAM COCO metrics.
>
> **References**:
>
> [10] Ashkboos, Saleh et al. “QuaRot: Outlier-Free 4-Bit Inference in Rotated LLMs.”
>
> [11] Tseng, Albert et al. “QuIP#: Even Better LLM Quantization with Hadamard Incoherence and Lattice Codebooks.” ArXiv abs/2402.04396
>
> [12] Shah, Jay et al. “FlashAttention-3: Fast and Accurate Attention with Asynchrony and Low-precision.” ArXiv abs/2407.08608 (2024): n. pag.

---

> ### Author Response · Authors · 2024-11-20
>
> We have also updated the paper in section A.10 and associated Table 21, which is an exact reproduction of AM-RADIO's training recipe, aside from using the smaller ViT-B and ViT-L models, and also applying PHI-S+MSE loss. This allows for a direct comparison to their results. Zero shot and dense perception metrics improve slightly over our SigLIP model with this change, while VLM metrics degrade. However, the ViT-L model is still generally better than AM-RADIO's ViT-H at half the parameter count. This demonstrates that the positive changes in quality are not attributable to the use of SigLIP, but rather the specific topic studied in the paper, which is distribution normalization across the teachers. Training a ViT-H was still outside of our budget for this study, however we believe that the trend in scalability was clear in both AM-RADIO and our study, and so a ViT-H would almost certainly improve further upon our produced ViT-L.

---

> ### Comment · Reviewer_Kdwv · 2024-11-26
> **Response to the rebuttal**
>
> First, I would like to thank the authors for the rebuttal and responding to my questions. After reading the rebuttal and other reviews, my main concern regarding the submission still remains. Regarding the scope of the work, I want to acknowledge the author's response. The proposed PHI-S normalization may or may not be applicable to other domains such as quantization. However, that would need extensive experimental results to determine the effectiveness (specifically for quantization that Hadamard based approaches already exist such as [10]) which will improve the quality of the work but requires major changes to the paper which is beyond the scope of this submission. As of this version, the submission introduces PHI-S as a normalization technique to improve multi-teacher distillation and specifically the AM-RADIO framework and provides experiments only for this setup. Given this, I expected stronger experimental results through head to head comparisons with AM-RADIO. Currently there are mixed results compared to the baseline on downstream tasks and multiple convoluted factors in experiments in various tables throughout the paper that makes it hard to have a fair understanding of the effectiveness of the method for this specific task. I also share some of the concerns raised by reviewer QTud on this front (i.e. the use of different teachers throughout the paper and the new results in Table 21 that shows the impact on downstream tasks, besides the use of different models between the proposed method and the baseline). Consequently, I would like to keep my rating.

---

> > ### Author Response · Authors · 2024-11-27
> >
> > Table 21 shows what happens when we exactly match AM-RADIO's recipe, with the following 2 changes (rows 4 and 5):
> > * We use ViT-B and ViT-L models due to compute constraints
> > * We use PHI-S + MSE as the feature distillation objective, instead of AM-RADIO's 0.9 * cos + 0.1 * smooth-l1
> >
> > In general, this puts us at a large disadvantage since we're comparing models that are 15% and 49% the size of AM-RADIO's ViT-H/16 model. Even considering the disadvantage, we observe the following results:
> > * Our ViT-B/16 model is *competitive* with AM-RADIO, aside from classification.
> > * Our ViT-L/16 model is *superior* to AM-RADIO on all tasks aside from classification and SAM-COCO.
> >
> > Regarding convoluted results, we've provided the following set of results to reviewer QTud, starting here: https://openreview.net/forum?id=rP7rghI7yt&noteId=QqqEehzha6
> >
> > What we find with these benchmark aggregation methods that don't rely on rank or fidelity is consistent with our main paper finding, which is that PHI-S remains the most effective method studied. For camera ready, we would likely replace Table 1 using ranks with the table present in this rebuttal (https://openreview.net/forum?id=rP7rghI7yt&noteId=J4cjteC2Rv). The conclusion is the same as the existing Table 1, however it also accounts for the magnitude of benchmark difference. The reason we originally chose not to use this version is because the interpretation complexity is higher without conveying more information than the average rank display.

---

### Author Response · Authors · 2024-11-22
**Reminder**

Dear reviewers,

As the discussion phase is ending soon, we request you to acknowledge our rebuttal and let us know if you have any further questions, otherwise please reassess the scores.

In addition to the individual responses we've provided to each reviewer, we'd like to draw attention to the following:

* We've introduced section "A.9.4 Fidelity" into the appendix, which provides a method of understanding the matching loss wrt the teacher distribution, and formulated in a way that allows aggregation across teachers to produce a single metric.
* We added a missing result for the ViT-L/14 model from UNIC to Table 20
* We exactly replicated AM-RADIO's training setup for an apples-apples comparison, only varying the loss function (e.g. applying PHI-S) and model size (keeping the ViT-B/L), but using their teacher set, partitioning strategy, and number of training steps. Results can be found in Table 21, which still demonstrate a consistent improvement using our proposed PHI-S method over AM-RADIO's baseline. This confirms that the major improvements come from the method we studied and not from a change in recipe.

Regards,
Authors

---

### Author Response · Authors · 2024-11-25
**Reminder 2**

Dear reviewers,

As the discussion phase is ending in about one day, we request you to acknowledge our rebuttal and let us know if you have any further questions, otherwise please reassess the scores. In response to reviewer QTud, we have further provided other ways to aggregate the benchmark suite to draw conclusions, which we believe further solidifies our introduction and use of the novel PHI Standardization (PHI-S) technique. These results can be found here: https://openreview.net/forum?id=rP7rghI7yt&noteId=QqqEehzha6

We believe that our work has importance and validity from a few viewpoints:
* **Theoretical** - We introduce the PHI Standardization technique and prove properties about it. PHI-S is agnostic to the application area, instead being applicable anywhere you have a normal multivariate distribution with arbitrary covariance. While the Hadamard matrix doesn't support truly arbitrary dimensions, as an extension to section A.5.4, it's entirely possible to embed an unsupported vector space into the nearest Hadamard-supported higher dimensionality (e.g. 3 -> 4), and apply the technique.
* **Applications** - Regardless of the employment of PHI-S, the study is internally consistent, and we find that using statistical normalization plays an important role in training agglomerative models. This could be seen as a confirmation of standardization being used by Theia and UNIC, but we also find that standardization (Global, Regular, PHI-S) works better than whitening (PCA, ZCA, HCA), which are new results not previously present in the literature. This reinforces an important point that agglomerative modeling needs to take the original distributions into account, and not all normalizations are created equal.
* **Applications** - We set a new state of the art on numerous benchmarks as compared against AM-RADIO, and do so with models that are 15% and 49% the size of AM-RADIO's ViT-H/16 model, only falling about 1% short of AM-RADIO's ability as a classifier, but beating it everywhere else.
* **Theoretical** - While not tied to any success with benchmarks in our study, the introduction and theoretical study of novel HCA provides insight into a whitening technique that has equal distortions on every dimension.
* **Theoretical** - Building upon Kessy et. al. [10] we show a unified understanding of what is happening across normalization techniques wrt distortion, which culminates in Figure 8 where it can be seen that the distortion of ZCA is phase shifted between PCA-W and HCA, and we also see how PHI-S's error spheroid relates to the phases of HCA.

Regards, Authors

[10] Agnan Kessy, Alex Lewin, and Korbinian Strimmer. Optimal Whitening and Decorrelation. The American Statistician, 72(4):309–314, October 2018. doi: 10.1080/00031305.2016.127.

---

### Meta-Review · Area_Chair_bt5G · 2024-12-16

**Metareview:**

The paper presents a modification of the AM-RADIO multi-task distillation framework. It proposes the normalization of the output distribution of the different teachers before distilling them into the students.  The main contribution of this work revolves around standardization of representations from the multiple teachers.  The experiments show that it is better on average than the compared methods.

Strengths:
- The motivation is solid and based on empirical observations from the teachers' activations.
- The results show that the proposal has slight improvements on average.
- Evaluation on several benchmarks.

However, the reviewers identified several shortcomings regarding the experimental results that do not support the paper's claims. In particular:
- Unfair comparisons given the difference in setups.
- Inconsistent teachers throughout the experiments.
- Limited results due to the narrow scope to AM-RADIO.
- The results are inconclusive due to the insignificant improvements.

Moreover, the reviewers mentioned that the idea of normalizing distributions within the AM-RADIO framework is not new and has been explored before. Thus, the main contribution is the limited experimental results.

Therefore, given the limited theoretical contributions and the inconsistency and inconclusiveness of the experimental results, I agree with the majority of the reviewers to reject the paper. While one reviewer had a positive evaluation, the strengths raised do not outweigh the identified issues with the proposal.

**Additional Comments On Reviewer Discussion:**

The authors updated the paper to address the reviewers' concerns and provided more details about the proposal.

After the rebuttal, reviewer Kdwv still had concerns regarding the scope of the work and the mixed results compared to the baselines in the paper. Moreover, reviewer QTud also raised concerns about the different teachers used in the method and unfair comparisons. Despite the authors’ responses and new metrics, the reviewer mentioned that the results are inconclusive. Similarly, reviewer JmDF mentioned that the results are inconclusive as there is no improvement from the proposal. This reviewer did not reply to the authors.

On the other hand, reviewer VJ1K had a positive evaluation of the paper and raised concerns regarding the motivation of the method and some experimental setup clarifications. In the exchange with the authors, this reviewer commented that the raised concerns were addressed during the rebuttal.

During the post-rebuttal discussion, the reviewers did not reply to my requests and discussion.

Given the issues raised by reviewers Kdwv, QTud, and JmDF, I weighed their concerns more heavily than the strengths raised by reviewer VJ1K. Moreover, since the concerns about the lack of evidence and inconclusive experimental results were prevalent across the three reviews, I gave them higher consideration. Additionally, two of these reviewers had exchanges with the authors and remained unconvinced after the exchange.

---

### Decision · Program_Chairs · 2025-01-22

Reject